# ALTERNATING DIFFERENTIATION FOR OPTIMIZATION LAYERS

**Haixiang Sun[1], Ye Shi[1],\* Jingya Wang[1], Hoang Duong Tuan[2], H. Vincent Poor[3], Dacheng Tao[4]**
ShanghaiTech University[1], University of Technology Sydney[2], Princeton University[3],
JD Explore Academy[4]
{sunhx, shiye, wangjingya}@shanghaitech.edu.cn,
Tuan.Hoang@uts.edu.au, poor@princeton.edu, dacheng.tao@gmail.com

## ABSTRACT

The idea of embedding optimization problems into deep neural networks as optimization layers to encode constraints and inductive priors has taken hold in recent years. Most existing methods focus on implicitly differentiating Karush–Kuhn–Tucker (KKT) conditions in a way that requires expensive computations on the Jacobian matrix, which can be slow and memory-intensive. In this paper, we developed a new framework, named Alternating Differentiation (Alt-Diff), that differentiates optimization problems (here, specifically in the form of convex optimization problems with polyhedral constraints) in a fast and recursive way. Alt-Diff decouples the differentiation procedure into a primal update and a dual update in an alternating way. Accordingly, Alt-Diff substantially decreases the dimensions of the Jacobian matrix especially for optimization with large-scale constraints and thus increases the computational speed of implicit differentiation. We show that the gradients obtained by Alt-Diff are consistent with those obtained by differentiating KKT conditions. In addition, we propose to truncate Alt-Diff to further accelerate the computational speed. Under some standard assumptions, we show that the truncation error of gradients is upper bounded by the same order of variables' estimation error. Therefore, Alt-Diff can be truncated to further increase computational speed without sacrificing much accuracy. A series of comprehensive experiments validate the superiority of Alt-Diff.

## 1 INTRODUCTION

Recent years have seen a variety of applications in machine learning that consider optimization as a tool for inference learning Belanger & McCallum (2016); Belanger et al. (2017); Diamond et al. (2017); Amos et al. (2017); Amos & Kolter (2017); Agrawal et al. (2019a). Embedding optimization problems as optimization layers in deep neural networks can capture useful inductive bias, such as domain-specific knowledge and priors. Unlike conventional neural networks, which are defined by an explicit formulation in each layer, optimization layers are defined implicitly by solving optimization problems. They can be treated as implicit functions where inputs are mapped to optimal solutions. However, training optimization layers together with explicit layers is not easy since explicit closed-form solutions typically do not exist for the optimization layers.

Generally, computing the gradients of the optimization layers can be classified into two main categories: differentiating the optimality conditions implicitly and applying unrolling methods. The ideas of differentiating optimality conditions have been used in bilevel optimization Kunisch & Pock (2013); Gould et al. (2016) and sensitivity analysis Bonnans & Shapiro (2013). Recently, OptNet Amos & Kolter (2017) and CvxpyLayer Agrawal et al. (2019a) have extended this method to optimization layers so as to enable end-to-end learning within the deep learning structure. However, these methods inevitably require expensive computation on the Jacobian matrix. Thus they are prone to instability and are often intractable, especially for large-scale optimization layers. Another direction to obtain the gradients of optimization layers is based on the unrolling methods Diamond et al. (2017); Zhang et al. (2023), where an iterative first-order gradient method is applied. However,

---

\*Corresponding author

they are memory-intensive since all the intermediate results have to be recorded. Besides, unrolling methods are not quite suitable to constrained optimization problems as expensive projection operators are needed.

In this paper, we aim to develop a new method that significantly increases the computational speed of the differentiation procedure for convex optimization problems with polyhedral constraints. Motivated by the scalability of the operator splitting method for optimization problem Glowinski & Le Tallec (1989); Stellato et al. (2020), we developed a new framework, namely Alternating Differentiation (Alt-Diff), that differentiates optimization layers in a fast and recursive way. Alt-Diff first decouples the constrained optimization problem into multiple subproblems based on the well-known alternating direction method of multipliers (ADMM) Boyd et al. (2011). Then, the differentiation operators for obtaining the derivatives of the primal and dual variables w.r.t the parameters are implemented on these subproblems in an alternating manner. Accordingly, Alt-Diff substantially decreases the dimensions of the Jacobian matrix, significantly increasing the computational speed of implicit differentiation, especially for large-scale problems. Unlike most existing methods that directly differentiate KKT conditions after obtaining the optimal solution for an optimization problem, Alt-Diff performs the forward and backward procedures simultaneously. Both the forward and backward procedures can be truncated without sacrificing much accuracy in terms of the gradients of optimization layers. Overall, our contributions are three-fold:

- We develop a new differentiation framework Alt-Diff that decouples the optimization layers in an alternating way. Alt-Diff significantly reduces the dimension of the KKT matrix, especially for optimization with large-scale constraints and thus increases the computational speed of implicit differentiation.

- We show that: 1) the gradient obtained by Alt-Diff are consistent with those obtained by differentiating the KKT conditions; 2) the truncation error of gradients is upper bounded by the same order of variables' estimation error given some standard assumptions. Therefore, Alt-Diff can be truncated to accelerate the computational speed without scarifying much accuracy.

- We conduct a series of experiments and show that Alt-Diff can achieve results comparable to state-of-the-art methods in much less time, especially for large-scale optimization problems. The fast performance of Alt-Diff comes from the dimension reduction of the KKT matrix and the truncated capability of Alt-Diff.

## 2 RELATED WORK

**Differentiation for optimization layers** Embedding optimization problems into deep learning architectures as optimization layers to integrate domain knowledge has received increasing attention in recent years Gould et al. (2016); Amos & Kolter (2017); Agrawal et al. (2019a); Gould et al. (2021). For example, Gould et al. (2016) collected some general techniques for differentiating argmin and argmax optimization problems based on the Implicit Function Theorem, while Amos & Kolter (2017) proposed a network architecture, namely OptNet, that integrates quadratic optimization into deep networks. This work was further extended for end-to-end learning with stochastic programming Donti et al. (2017). Recently, an efficient differentiable quadratic optimization layer Butler & Kwon (2022) based on ADMM was proposed to accelerate the differentiation procedure. However, these methods are only restricted to handling the differentiation of quadratic problems. For more general convex optimization, Agrawal et al. (2019b) proposed CvxpyLayer that differentiates through disciplined convex programs and some sparse operations and LSQR Paige & Saunders (1982) to speed up the differentiation procedure. Although these have been worthwhile attempts to accelerate the differentiation procedure, they still largely focus on implicitly differentiating the KKT conditions in a way that requires expensive operations on a large-scale Jacobian matrix. Recently, a differentiation solver named JaxOpt Blondel et al. (2021) has also been put forward that is based on an implicit automatic differentiation mechanism under the Jax framework.

**Unrolling methods** Another direction for differentiating optimization problems is the unrolling methods Domke (2012); Monga et al. (2021). These approximate the objective function with a first-order gradient method and then incorporate the gradient of the inner loop optimization into the training procedures Belanger & McCallum (2016); Belanger et al. (2017); Amos et al. (2017); Metz

et al. (2019). This is an iterative procedure, which is usually truncated to a fixed number of iterations. Although unrolling methods are easy to implement, most of their applications are limited to unconstrained problems. If constraints are added, the unrolling solutions have to be projected into the feasible region, significantly increasing the computational burdens. By contrast, Alt-Diff only requires a very simple operation that projects the slack variable to the nonnegative orthant. This substantially improves the efficiency of subsequent updates and reduces the method's computational complexity.

**Implicit models** There has been growing interest in implicit models in recent years J. Z. Kolter & Johnson (2020); Zhang et al. (2020); Bolte et al. (2021). Implicit layers replace the traditional feed-forward layers of neural networks with fixed point iterations to compute inferences. They have been responsible for substantial advances in many applications, including Neural ODE Chen et al. (2018); Dupont et al. (2019), Deep Equilibrium Models Bai et al. (2019; 2020); Gurumurthy et al. (2021), nonconvex optimization problems Wang et al. (2019) and implicit 3D surface layers Michalkiewicz et al. (2019); Park et al. (2019), etc. Implicit layers have a similar computational complexity to optimization layers as they also requires solving a costly Jacobian-based equation based on the Implicit Function Theorem. Recently, Fung et al. (2021) proposed a Jacobian-free backpropagation method to accelerate the speed of training implicit layers. However, this method is not suitable for the optimization layers with complex constraints. In terms of training implicit models, Geng et al. (2021) also proposed a novel gradient estimate called phantom gradient which relies on fixed-point unrolling and a Neumann series to provide a new update direction; computation of precise gradient is forgone. Implicit models have also been extended to more complex learning frameworks, such as attention mechanisms Geng et al. (2020) and Graph Neural Networks Gu et al. (2020).

## 3 PRELIMINARY: DIFFERENTIABLE OPTIMIZATION LAYERS

We consider a parameterized convex optimization problems with polyhedral constraints:

$$
\begin{aligned}
\min_{x} \quad & f(x; \theta) \\
s.t. \quad & x \in \mathcal{C}(\theta)
\end{aligned}
\tag{1}
$$

where $x \in \mathbb{R}^n$ is the decision variable, the objective function $f : \mathbb{R}^n \to \mathbb{R}$ is convex and the constraint $\mathcal{C}(\theta) := \{x | Ax = b, \quad Gx \leq h\}$ is a polyhedron. For simplification, we use $\theta$ to collect the parameters in the objective function and constraints. For any given $\theta$, a solution of optimization problem (1) is $x^\star \in \mathbb{R}^n$ that minimizes $f(x; \theta)$ while satisfying the constraints, i.e. $x^\star \in \mathcal{C}(\theta)$. Then the optimization problem (1) can be viewed as a mapping that maps the parameters $\theta$ to the solution $x^\star$. Here, we focus on convex optimization with affine constraints due to its wide applications in control systems Guo & Wang (2010), signal processing Mattingley & Boyd (2010), communication networks Luong et al. (2017); Bui et al. (2017), etc.

Since optimization problems can capture well-defined specific domain knowledge in a model-driven way, embedding such optimization problems into neural networks within an end-to-end learning framework can simultaneously leverage the strengths of both the model-driven and the data-driven methods. Unlike conventional neural networks, which are defined through explicit expressions of each layer, an optimization layer embedded in optimization problem (1) is defined as follows:

**Definition 3.1.** (*Optimization Layer*) A layer in a neural network is defined as an optimization layer if its input is the optimization parameters $\theta \in \mathbb{R}^m$ and its output $x^\star \in \mathbb{R}^n$ is the solution of the optimization problem (1).

The optimization layer can be treated as an implicit function $\mathcal{F} : \mathbb{R}^n \times \mathbb{R}^m \to \mathbb{R}^n$ with $\mathcal{F}(x^\star, \theta) = \mathbf{0}$. In the deep learning architecture, optimization layers are implemented together with other explicit layers in an end-to-end framework. During the training procedure, the chain rule is used to back propagate the gradient through the optimization layer. Given a loss function $\mathcal{R}$, the derivative of the loss w.r.t the parameter $\theta$ of the optimization layer is

$$
\frac{\partial \mathcal{R}}{\partial \theta} = \frac{\partial \mathcal{R}}{\partial x^\star} \frac{\partial x^\star}{\partial \theta}.
\tag{2}
$$

Obviously, the derivative $\frac{\partial \mathcal{R}}{\partial x^\star}$ can be easily obtained by automatic differentiation techniques on explicit layers, such as fully connected layers and convolutional layers. However, since explicit

closed-form solutions typically do not exist for optimization layers, this brings additional computational difficulties during implementation. Recent work, such as OptNet Amos & Kolter (2017) and CvxpyLayer Agrawal et al. (2019a) have shown that the Jacobian $\frac{\partial x^\star}{\partial \theta}$ can be derived by differentiating the KKT conditions of the optimization problem based on the Implicit Function Theorem Krantz & Parks (2002). This is briefly recalled in Lemma 3.2.

**Lemma 3.2.** *(Implicit Function Theorem) Let $\mathcal{F}(x, \theta)$ denote a continuously differentiable function with $\mathcal{F}(x^\star, \theta) = \mathbf{0}$. Suppose the Jacobian of $\mathcal{F}(x, \theta)$ is invertible at $(x^\star; \theta)$, then the derivative of the solution $x^\star$ with respect to $\theta$ is*

$$\frac{\partial x^\star}{\partial \theta} = - \left[ \mathcal{J}_{\mathcal{F};x} \right]^{-1} \mathcal{J}_{\mathcal{F};\theta}, \tag{3}$$

*where $\mathcal{J}_{\mathcal{F};x} := \nabla_x \mathcal{F}(x^\star, \theta)$ and $\mathcal{J}_{\mathcal{F};\theta} := \nabla_\theta \mathcal{F}(x^\star, \theta)$ are respectively the Jacobian of $\mathcal{F}(x, \theta)$ w.r.t. $x$ and $\theta$.*

It is worth noting that the differentiation procedure needs to calculate the optimal value $x^\star$ in the forward pass and involves solving a linear system with Jacobian matrix $\mathcal{J}_{\mathcal{F};x}$ in the backward pass. Both the forward and backward pass are generally very computationally expensive, especially for large-scale problems. This means that differentiating KKT conditions directly is not scalable to large optimization problems. Existing solvers have made some attempts to alleviate this issue. Specifically, CvxpyLayer adopted an LSQR technique to accelerate implicit differentiation for sparse optimization problems. However, the method is not necessarily efficient for more general cases, which may not be sparse. Although OptNet uses a primal-dual interior point method in the forward pass and its backward pass can be very easy but OptNet is suitable only for quadratic optimization problems. In this paper, our main target is to develop a new method that can increase the computational speed of the differentiation procedure especially for large-scale constrained optimization problems.

## 4 ALTERNATING DIFFERENTIATION FOR OPTIMIZATION LAYERS

This section provides the details of the Alt-Diff algorithm for optimization layers. Alt-Diff decomposes the differentiation of a large-scale KKT system into a smaller problems that are solved in a primal-dual alternating way (see Section 4.1). Alt-Diff reduces the computational complexity of the model and significantly improves the computational efficiency of the optimization layers. In section 4.2, we theoretically analyze the truncated capability of Alt-Diff. Notably, Alt-Diff can be truncated for inexact solutions to further increase computational speeds without sacrificing much accuracy.

### 4.1 ALTERNATING DIFFERENTIATION

Motivated by the scalability of the operator splitting method Glowinski & Le Tallec (1989); Stellato et al. (2020), Alt-Diff first decouples a constrained optimization problem into multiple subproblems based on ADMM. Each splitted operator is then differentiated to establish the derivatives of the primal and dual variables w.r.t the parameters in an alternating fashion. The augmented Lagrange function of problem (1) with a quadratic penalty term is:

$$\begin{aligned} \max_{\lambda, \nu} \min_{x, s \geq 0} &\mathcal{L}(x, s, \lambda, \nu; \theta) \\ &= f(x; \theta) + \langle \lambda, Ax - b \rangle + \langle \nu, Gx + s - h \rangle + \frac{\rho}{2}(\|Ax - b\|^2 + \|Gx + s - h\|^2), \end{aligned} \tag{4}$$

where $s \geq 0$ is a non-negative slack variable, and $\rho > 0$ is a hyperparameter associated with the penalty term. Accordingly, the following ADMM procedures are used to alternatively update the primary, slack and dual variables:

$$\begin{cases} x_{k+1} = \arg \min_{x} \mathcal{L}(x, s_k, \lambda_k, \nu_k; \theta), & \text{(5a)} \\[2mm] s_{k+1} = \arg \min_{s \geq 0} \mathcal{L}(x_{k+1}, s, \lambda_k, \nu_k; \theta), & \text{(5b)} \\[2mm] \lambda_{k+1} = \lambda_k + \rho(Ax_{k+1} - b), & \text{(5c)} \\[2mm] \nu_{k+1} = \nu_k + \rho(Gx_{k+1} + s_{k+1} - h). & \text{(5d)} \end{cases}$$

Note that the primal variable $x_{k+1}$ is updated by solving an unconstrained optimization problem. The update of the slack variable $s_{k+1}$ is easily obtained by a closed-form solution via a Rectified Linear Unit (ReLU) as

$$s_{k+1} = \textbf{ReLU}\left(-\frac{1}{\rho}\nu_k - (Gx_{k+1} - h)\right). \tag{6}$$

The differentiations for the slack and dual variables are trivial and can be done using an automatic differentiation technique. Therefore, the computational difficulty is now concentrated around differentiating the primal variables, which can be done using the Implicit Function Theorem. Applying a differentiation technique to procedure (5) leads to:

$$\begin{cases} \dfrac{\partial x_{k+1}}{\partial \theta} = -\left(\nabla_x^2 \mathcal{L}(x_{k+1})\right)^{-1} \nabla_{x,\theta}\mathcal{L}(x_{k+1}), & \text{(7a)} \\[3mm] \dfrac{\partial s_{k+1}}{\partial \theta} = -\dfrac{1}{\rho}\textbf{sgn}(s_{k+1}) \cdot \mathbf{1}^T \odot \left(\dfrac{\partial \nu_k}{\partial \theta} + \rho\dfrac{\partial(Gx_{k+1} - h)}{\partial \theta}\right), & \text{(7b)} \\[3mm] \dfrac{\partial \lambda_{k+1}}{\partial \theta} = \dfrac{\partial \lambda_k}{\partial \theta} + \rho\dfrac{\partial(Ax_{k+1} - b)}{\partial \theta}, & \text{(7c)} \\[3mm] \dfrac{\partial \nu_{k+1}}{\partial \theta} = \dfrac{\partial \nu_k}{\partial \theta} + \rho\dfrac{\partial(Gx_{k+1} + s_{k+1} - h)}{\partial \theta}, & \text{(7d)} \end{cases}$$

where $\nabla_x^2 \mathcal{L}(x_{k+1}) = \nabla_x^2 f(x_{k+1})^T + \rho A^T A + \rho G^T G$, $\odot$ represents Hadamard production, $\textbf{sgn}(s_{k+1})$ denotes a function such that $\textbf{sgn}(s_{k+1}^i) = 1$ if $s_{k+1}^i \geq 0$ and $\textbf{sgn}(s_{k+1}^i) = 0$ vice versa.

We summarize the procedure of Alt-Diff in Algorithm 1 and provide a framework of Alt-Diff in Appendix A. The convergence behavior of Alt-Diff can be inspected by checking $\|x_{k+1}-x_k\|/\|x_k\| < \epsilon$, where $\epsilon$ is a predefined threshold.

Notably, Alt-Diff is somehow similar to unrolling methods as in Foo et al. (2007); Domke (2012). However, these unrolling methods were designed for unconstrained optimization. If constraints are added, the unrolling solutions have to be projected into a feasible region. Generally this pro-

---

**Algorithm 1** Alt-Diff

**Parameter**: $\theta$ from the previous layer
Initialize slack variable and dual variables
    **while** $\|x_{k+1} - x_k\|/\|x_k\| \geq \epsilon$ **do**
        **Forward update** by (5)
        Primal update $\frac{\partial x_{k+1}}{\partial \theta}$ by (7a)
        Slack update $\frac{\partial s_{k+1}}{\partial \theta}$ by (7b)
        Dual update $\frac{\partial \lambda_{k+1}}{\partial \theta}$ and $\frac{\partial \nu_{k+1}}{\partial \theta}$ by (7c) and (7d)
        $k := k + 1$
    **end while**
    **return** $x^\star$ and its gradient $\frac{\partial x^\star}{\partial \theta}$

---

jector operators are very computationally expensive. By contrast, Alt-Diff can decouple constraints from the optimization and only involves a very simple operation that projects the slack variable $s$ to the nonnegative orthant $s \geq 0$. This significantly improves the efficiency of subsequent updates and reduces the overall computational complexity of Alt-Diff. Besides, conventional unrolling methods usually need more memory consumption as all the intermediate computational results have to be recorded. However, when updating (7) continuously, Alt-Diff does not need to save the results of the previous round. Instead, it only saves the results of the last round, that is, the previous $\frac{\partial x_k}{\partial \theta}$ is replaced by the $\frac{\partial x_{k+1}}{\partial \theta}$ iteratively. Besides, we show that the gradient obtained by Alt-Diff is consistent with the one obtained by the gradient of the optimality conditions implicitly. A detailed proof as well as the procedure for differentiating the KKT conditions is presented in the Appendix E.

For more clarity, we take several optimization layers for examples to show the implementation of Alt-Diff, including Quadratic Layer Amos & Kolter (2017), constrained Sparsemax Layer Malaviya et al. (2018) and constrained Softmax Layer Martins & Astudillo (2016). These optimization layers were proposed to integrate specific domain knowledge in the learning procedure. Detailed derivations for the primal differentiation procedure (7a) are provided in the Appendix B.2.

In the forward pass, the unconstrained optimization problem (5a) can be easily solved by Newton's methods where the inverse of Hessian, i.e. $\left(\nabla_x^2 \mathcal{L}(x_{k+1})\right)^{-1}$ is needed. It can be directly used in the backward pass (7a) to accelerate the computation after the forward pass. The computation is even more efficient in the case of quadratic programmings, where $\left(\nabla_x^2 \mathcal{L}(x_{k+1})\right)^{-1}$ is constant and only

needs to be computed once. More details are referred to Appendix B.1. As Alt-Diff decouples the objective function and constraints, it is obviously that the dimension of Hessian matrix $\nabla_x^2 \mathcal{L}(x_{k+1})$ is reduced to the number of variables $n$. Therefore, the complexity for Alt-Diff becomes $\mathcal{O}(n^3)$, which differs from directly differentiating the KKT conditions with complexity of $\mathcal{O}((n + n_c)^3)$. This also validates the superiority of Alt-Diff for optimization problem with many constraints. To further accelerate Alt-Diff, we propose to truncate its iterations and analyze the effects of truncated Alt-Diff.

## 4.2 TRUNCATED CAPABILITY OF ALT-DIFF

As shown in recent work Fung et al. (2021); Geng et al. (2021), exact gradients are not necessary during the learning process of neural networks. Therefore, we can truncate the iterative procedure of Alt-Diff given some threshold values $\epsilon$. The truncation in existing methods (OptNet and CvxpyLayers) that are based on interior point methods often does not offer significant improvements. This is because the interior point methods often converge fast whereas the computational bottleneck comes from the high dimension of the KKT matrix instead of the number of iterations of the interior points methods. However, the iteration truncation can benefit Alt-Diff more as the iteration of Alt-Diff is more related to the tolerance due to the sublinear convergence of ADMM. Our simulation results in Section 5.1 and Appendix F.1 also verify this phenomenon. Next, we give theoretical analysis to show the influence of truncated Alt-Diff.

**Assumption A** *(L-smooth)* The first order derivatives of loss function $\mathcal{R}$ and the second derivatives of the augmented Lagrange function $\mathcal{L}$ are $L$-Lipschitz. For $\forall x_1, x_2 \in \mathbb{R}^n$,

$$\|\nabla_x \mathcal{R}(\theta; x_1) - \nabla_x \mathcal{R}(\theta; x_2)\| \le L_1 \|x_1 - x_2\|, \tag{8a}$$

$$\|\nabla_x^2 f(x_1) - \nabla_x^2 f(x_2)\| \le L_2 \|x_1 - x_2\|, \tag{8b}$$

$$\|\nabla \mathcal{L}_{x,\theta}(x_1) - \nabla \mathcal{L}_{x,\theta}(x_2)\| \le L_3 \|x_1 - x_2\|. \tag{8c}$$

**Assumption B** *(Bound of gradients)* The first and second order derivative of function $\mathcal{L}$ and $\mathcal{R}$ are bounded as follows:

$$\nabla_x \mathcal{R}(\theta; x) \preceq \mu_1 I, \qquad \nabla_x^2 \mathcal{L}(x) \succeq \mu_2 I, \qquad \nabla_{x,\theta} \mathcal{L}(x) \preceq \mu_3 I, \tag{9}$$

where $\mu_1$, $\mu_2$ and $\mu_3$ are positive constants. The above inequalities hold for $\forall x \in \mathbb{R}^n$.

**Assumption C** *(Nonsingular Hessian)* Function $f(x)$ is twice differentiable. The Hessian matrix of augmented Lagrange function $\mathcal{L}$ is invertiable at any point $x \in \mathbb{R}^n$.

**Theorem 4.1.** *(Error of gradient obtained by truncated Alt-Diff) Suppose $x_k$ is the truncated solution at the $k$-th iteration, the error between the gradient $\frac{\partial x_k}{\partial \theta}$ obtained by the truncated Alt-Diff and the real gradient $\frac{\partial x^\star}{\partial \theta}$ is bounded as follows,*

$$\left\| \frac{\partial x_k}{\partial \theta} - \frac{\partial x^\star}{\partial \theta} \right\| \le C_1 \|x_k - x^\star\|, \tag{10}$$

*where $C_1 = \frac{L_3}{\mu_2} + \frac{\mu_3 L_2}{\mu_2^2}$ is a constant.*

Please refer to Appendix C for our proof. The theorem has shown that the primal differentiation (7a) obtained by Alt-Diff has the same order of error brought by the truncated iterations of (5a). Besides, this theorem also illustrates the convergence of Alt-Diff. By the convergence of ADMM itself, $x_k$ will converge to $x^\star$. The convergence of the Jacobian is established accordingly and the results from the computational perspective are shown in Appendix E.2. Moreover, we can derive the following corollary with respect to the general loss function $\mathcal{R}$ accordingly.

**Corollary 4.2.** *(Error of the inexact gradient in terms of loss function) Followed by Theorem 4.1, the error of the gradient w.r.t. $\theta$ in loss function $\mathcal{R}$ is bounded as follows:*

$$\|\nabla \mathcal{R}(\theta; x_k) - \nabla \mathcal{R}(\theta; x^\star)\| \le C_2 \|x_k - x^\star\|, \tag{11}$$

*where $C_2 = L_1 + \frac{\mu_3 L_1 + \mu_1 L_3}{\mu_2} + \frac{\mu_1 \mu_3 L_2}{\mu_2^2}$ is a constant.*

Please refer to Appendix D for our proof. Truncating the iterative process in advance will result in fewer steps and higher computational efficiency without sacrificing much accuracy. In the next section, we will show the truncated capability of Alt-Diff by several numerical experiments.

## 5 EXPERIMENTAL RESULTS

In this section, we evaluate Alt-Diff over a series of experiments to demonstrate its performance in terms of computational speed as well as accuracy. First, we implemented Alt-Diff over several optimization layers, including the sparse and dense constrained quadratic layers and constrained softmax layers. In addition, we applied the Alt-Diff to the real-world task of energy generation scheduling under a predict-then-optimize framework. Moreover, we also tested Alt-Diff with a quadratic layer in an image classification task on the MNIST dataset. To verify the truncated capability of Alt-Diff, we also implemented Alt-Diff with different values of tolerance. All the experiments were implemented on a Core Intel(R) i7-10700 CPU @ 2.90GHz with 16 GB of memory. Our source code for these experiments is available at https://github.com/HxSun08/Alt-Diff.

### 5.1 NUMERICAL EXPERIMENTS ON SEVERAL OPTIMIZATION LAYERS

In this section, we compare Alt-Diff with OptNet and CvxpyLayer on several optimization layers, including the constrained Sparsemax layer, the dense Quadratic layer and the constrained Softmax layer which are referred to typical cases of sparse, dense quadratic problems and problems with general convex objective function, respectively.

In the dense quadratic layer, with the objective function as $f(x) = \frac{1}{2}x^T P x + q^T x$, the parameters $P, q, A, b, G, h$ were randomly generated from the same random seed with $P \succeq 0$. The tolerance $\epsilon$ is set as $10^{-3}$ for OptNet, CvxpyLayer and Alt-Diff. As the parameters in optimization problems are generated randomly, we compared Alt-Diff with the "dense" mode in CvxpyLayer. All the numerical experiments were executed 5 times with the average times reported as the results in Table 1. It can be seen that OptNet runs much faster than CvxpyLayer for the dense quadratic layer, while our Alt-Diff outperforms both the two counterparts. Additionally, the superiority of Alt-Diff becomes more evident with the increase of problem size. We also plot the trend of Jacobian $\frac{\partial x_k}{\partial b}$ with iterations in primal update (7a) in Figure 1. As shown in Theorem 4.1, the gradient obtained by Alt-Diff can gradually converge to the results obtained by KKT derivative as the number of iterations increases. The results in Figure 1 also validate this theorem.

Table 1: Comparison of running time (s) and cosine distances of gradients in dense quadratic layers with tolerance $\epsilon = 10^{-3}$.

|  | Tiny | Small | Medium | Large |
|---|---|---|---|---|
| Num of variable $n$ | 1500 | 3000 | 5000 | 10000 |
| Num of ineq. $m$ | 500 | 1000 | 2000 | 5000 |
| Num of eq. $p$ | 200 | 500 | 1000 | 2000 |
| Num of elements | $4.84 \times 10^6$ | $2.03 \times 10^7$ | $6.4 \times 10^7$ | $2.89 \times 10^8$ |
| **OptNet (total)** | 0.81 | 9.48 | 40.72 | 317.78 |
| Forward | 0.77 | 9.29 | 40.19 | 315.21 |
| Backward | 0.04 | 0.19 | 0.53 | 2.57 |
| **CvxpyLayer (total)** | 25.34 | 212.86 | 1056.18 | - |
| Initialization | 1.86 | 10.51 | 38.73 | - |
| Canonicalization | 0.37 | 1.47 | 4.40 | - |
| Forward | 14.00 | 117.06 | 572.24 | - |
| Backward | 9.11 | 83.82 | 440.80 | - |
| **Alt-Diff (total)** | **0.71** | **5.75** | **20.20** | **154.52** |
| Inversion | 0.11 | 0.91 | 3.03 | 21.12 |
| Forward and backward | 0.60 | 4.84 | 17.17 | 133.40 |
| Cosine Dist. | 0.999 | 0.999 | 0.999 | 0.998 |

"-" represents that the solver cannot generate the gradients.

Further, we conduct experiments to compare the truncated performance among Alt-Diff, OptNet and CvxpyLayer under different tolerance levels $\epsilon$. As shown in Table 2, we can find that the truncated operation will not significantly improve the computational speed for existing methods but does have significant improvements for Alt-Diff. The detailed results of other optimization layers are provided in Appendix F.1.

Table 2: Comparison of running time (s) under different tolerances for a dense quadratic layer with $n = 3000$, $m = 1000$ and $p = 500$.

| Tolerance $\epsilon$ | $10^{-1}$ | $10^{-2}$ | $10^{-3}$ | $10^{-4}$ | $10^{-5}$ |
|---|---|---|---|---|---|
| OptNet | 8.58 | 9.32 | 9.48 | 9.59 | 10.13 |
| CvxpyLayer | 212.34 | 216.66 | 212.86 | 215.75 | 213.08 |
| Alt-Diff | 1.18 | 3.50 | 5.75 | 8.04 | 10.76 |

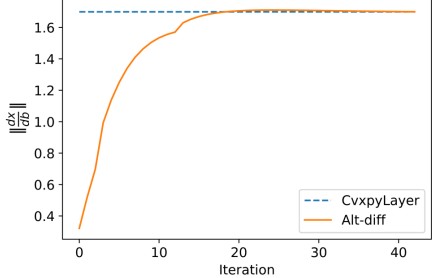

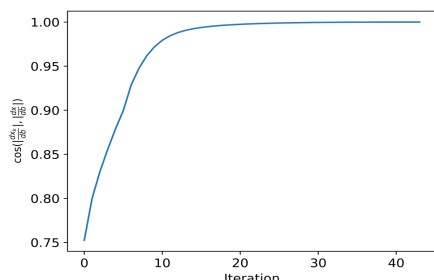

(a) The norm variation trends. (Blue dotted line is the gradient $\frac{\partial x^\star}{\partial b}$ obtained by CvxpyLayer).

(b) The cosine distance of the obtained gradient between Alt-Diff and CvxpyLayer.

Figure 1: The variation of primal variable gradient of dense quadratic layers. (The threshold of iteration is set as $10^{-3}$)

## 5.2 ENERGY GENERATION SCHEDULING

In order to test the accuracy of sparse layers, we applied the Alt-Diff to the real-world task of energy generation scheduling Donti et al. (2017) under a predict-then-optimize framework Elmachtoub & Grigas (2021); Mandi et al. (2020); Mandi & Guns (2020). Predict-then-optimize is an end-to-end learning model, in which some unknown variables are first predicted by some machine learning methods and are then successively optimized. The main idea of predict-then-optimize is to use the optimization loss (12) to guide the prediction, rather than use the prediction loss as in the normal learning style.

$$L(\hat{\theta}, \theta) = \frac{1}{2} \sum_{i=1}^{m} (x_i^\star(\hat{\theta}) - x_i^\star(\theta))^2 \tag{12}$$

We consider the energy generation scheduling task based on the actual power demand for electricity in a certain region web. In this setting, a power system operator must decide the electricity generation to schedule for the next 24 hours based on some historical electricity demand information. In this paper, we used the hourly electricity demand data over the past 72 hours to predict the real power demand in the next 24 hours. The predicted electricity demand was then input into the following optimization problem to schedule the power generation:

$$\min_{x_k} \quad \sum_{k=1}^{24} \|x_k - P_{d_k}\|^2$$
$$s.t. \quad |x_{k+1} - x_k| \leq r, \quad k = 1, 2, \cdots, 23 \tag{13}$$

where $P_{d_k}$ and $x_k$ denote the power demand and power generation at time slot $k$ respectively. Due to physical limitations, the variation of power generation during a single time slot is not allowed to exceed a certain threshold value $r$. During training procedures, we use a neural network with two hidden layers to predict the electricity demand of the next 24 hours based on the data of the previous 72 hours. All the $P_{d_k}$ have been normalized into the $[0, 100]$ interval.

The optimization problem in this task can be treated like the optimization layer previously considered. As shown in Table 4, since the constraints are sparse, OptNet runs much slower than the "lsqr" mode in CvxpyLayer. Therefore, we respectively implemented Alt-Diff and CvxpyLayer to obtain

the gradient of power generation with respect to the power demand. During the training procedure, we used the Adam optimizer Kingma & Ba (2014) to update the parameter. Once complete, we compared the results obtained from CvxpyLayer (with tolerance $10^{-3}$) and those from Alt-Diff under different levels of truncated thresholds, varying from $10^{-1}$, $10^{-2}$ to $10^{-3}$. The results are shown in Figure 2. We can see that the losses for CvxpyLayer and Alt-Diff with different truncated threshold values are almost the same, but the running time of Alt-Diff is much smaller than CvxpyLayer especially in the truncated cases.

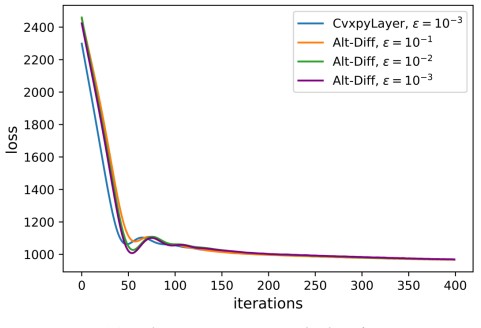

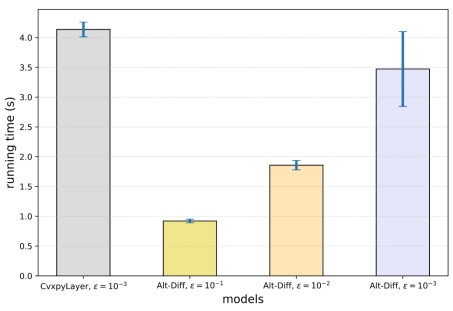

(a) The convergence behaviors.       (b) The average running time.

Figure 2: The experimental results of Alt-diff and CvxpyLayer for energy generation scheduling.

### 5.3 IMAGE CLASSIFICATION

We follow similar experimental settings in OptNet Amos & Kolter (2017) to embed the optimization layer into the neural networks for image classification task on MNIST dataset. A layer in the neural networks is replaced by a dense quadratic optimization layer. Then we compare the computational speed and accuracy between Alt-Diff and OptNet. As the experiment results shown in Appendix F.2, we can see Alt-Diff runs much faster and yields similar accuracy compared to OptNet. Since OptNet runs much faster than CvxpyLayer in dense quadratic layers (as shown in Section 5.1), therefore we only compare Alt-Diff with OptNet here. More details are provided in the Appendix F.2 due to page limitation.

## 6 CONCLUSION

In this paper, we have proposed Alt-Diff for computationally efficient differentiation on optimization layers associated with convex objective functions and polyhedral constraints. Alt-Diff differentiates the optimization layers in a fast and recursive way. Unlike differentiation on the KKT conditions of optimization problem, Alt-Diff decouples the differentiation procedure into a primal update and a dual update by an alternating way. Accordingly, Alt-Diff substantially decreases the dimensions of the Jacobian matrix especially for large-scale constrained optimization problems and thus increases the computational speed. We have also showed the convergence of Alt-Diff and the truncated error of Alt-Diff given some general assumptions. Notably, we have also shown that the iteration truncation can accelerate Alt-Diff more than existing methods. Comprehensive experiments have demonstrated the efficiency of Alt-Diff compared to the state-of-the-art. Apart from the superiority of Alt-Diff, we also want to highlight its potential drawbacks. One issue is that the forward pass of Alt-Diff is based on ADMM, which does not necessarily always outperform the interior point methods. In addition, Alt-Diff is currently designed only for specific optimization layers in the form of convex objective functions with polyhedral constraints. Extending Alt-Diff to more general convex optimization layers and nonconvex optimization layers is under consideration in our future work.

ACKNOWLEDGMENTS

This work was supported by the Shanghai Sailing Program (22YF1428800, 21YF1429400), Shanghai Local college capacity building program (23010503100), and Shanghai Frontiers Science Center of Humancentered Artificial Intelligence (ShangHAI).

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

# Appendix

We present some details for the proposed Alt-Diff algorithm, including the framework of Alt-Diff, the derivation of some specific layers, the proof of theorems, detailed experimental results and implementation notes.

## A  FRAMEWORK OF ALT-DIFF

The Alt-Diff framework for optimization layers within an end-to-end learning architecture is illustrated in Figure 3.

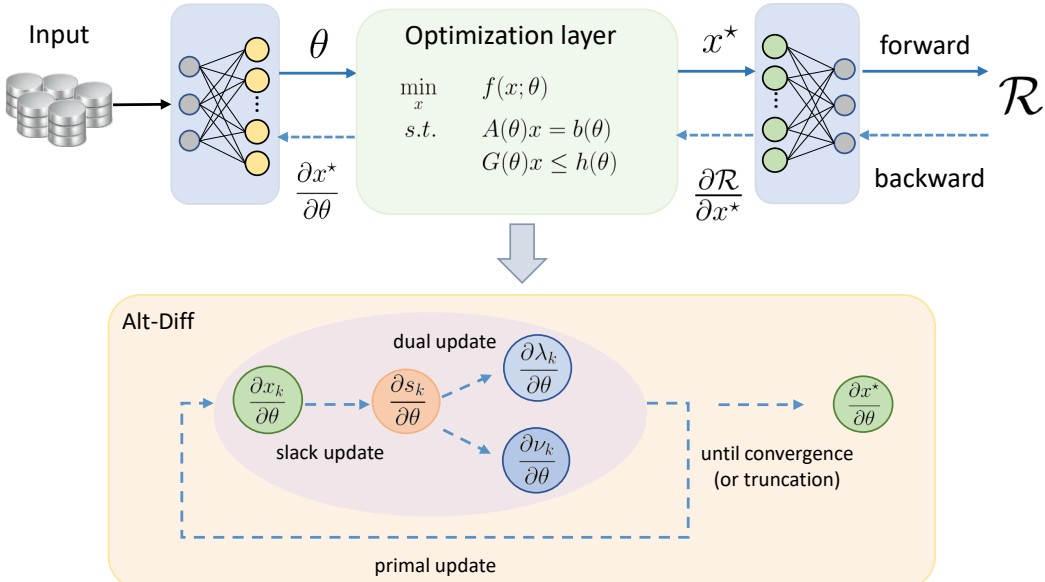

Figure 3: The model architecture of Alt-Diff for optimization layers.

## B  PRIMAL DIFFERENTIATION

### B.1  INHERITANCE OF HESSIAN MATRIX

According to the first-order optimality condition in forward pass (5a), we have

$$\nabla \mathcal{L}(x_{k+1}) = \nabla f(x_{k+1}) + A^T \lambda_k + G^T \nu_k + \rho A^T (A x_{k+1} - b) + \rho G^T (G x_{k+1} + s_k - h) = 0. \tag{14}$$

Generally, the equation (14) can be efficiently solved by iterative methods such as Newton's methods as follows:

$$
\begin{aligned}
x_{k+1}^i &= x_{k+1}^{i-1} - \alpha \left( \nabla_x^2 \mathcal{L}\left( x_{k+1}^{i-1} \right) \right)^{-1} \nabla \mathcal{L}\left( x_{k+1}^{i-1} \right) \\
&= x_{k+1}^{i-1} - \alpha \left( \nabla_x^2 f\left( x_{k+1}^{i-1} \right) + \rho A^T A + \rho G^T G \right)^{-1} \nabla \mathcal{L}\left( x_{k+1}^{i-1} \right),
\end{aligned}
\tag{15}
$$

where $x_{k+1}^i$ denotes the value of $x_{k+1}$ at the $i$-th Newton's iteration and $\alpha$ is the step size. Therefore, when $x_{k+1}^i$ converges at its optimal value $x_{k+1}$, i.e., after the forward pass, the inverse of Hessian $\left( \nabla_x^2 \mathcal{L}(x_{k+1}) \right)^{-1}$ can be directly used to accelerate the computation of the backward pass (7a). Especially, when $f(x)$ is quadratic, i.e. $f(x) = \frac{1}{2} x^T P x + q^T x$, and $P$ is a symmetric matrix, the inverse of Hessian becomes constant as:

$$\left(\nabla_x^2 \mathcal{L}(x_{k+1})\right)^{-1} = \left(P + \rho A^T A + \rho G^T G\right)^{-1}. \tag{16}$$

## B.2 SPECIAL LAYERS

We take Quadratic Layer Amos & Kolter (2017), constrained Sparsemax Layer Malaviya et al. (2018) and constrained Softmax Layer Martins & Astudillo (2016) as special cases in order to show the computation of backward pass (7). The details of these layers are provided in the first two columns of Table 3. The differentiation procedure (7b) - (7d) for these layers are exactly the same. The only difference relies on the primal differentiation procedure (7a), which is listed in the last column of Table 3.

Table 3: The primal differentiation procedure of Alt-Diff for several optimization layers.

| Layers | Optimization problems | Primal differentiation (7a) |
|---|---|---|
| Constrained Sparsemax Layer | $\min_x \|x - y\|_2^2 \ s.t. \ 1^T x = 1, \ 0 \le x \le u$ | $\frac{\partial x_{k+1}}{\partial \theta} = -\left((2+2\rho)I + \rho \mathbf{1} \cdot \mathbf{1}^T\right)^{-1} \mathcal{L}_{x\theta}(x_{k+1})$ |
| Quadratic Layer | $\min_x \frac{1}{2} x^T P x + q^T x \ s.t. \ Ax = b, \ Gx \le h$ | $\frac{\partial x_{k+1}}{\partial \theta} = -\left(P + \rho A^T A + \rho G^T G\right)^{-1} \mathcal{L}_{x\theta}(x_{k+1})$ |
| Constrained Softmax Layer | $\min_x \ -y^T x + H(x) \ s.t. \ 1^T x = 1, \ 0 \le x \le u$ | $\frac{\partial x_{k+1}}{\partial \theta} = -\left(\mathbf{diag}^{-1}(x) + 2\rho I + \rho \mathbf{1} \cdot \mathbf{1}^T\right)^{-1} \mathcal{L}_{x\theta}(x_{k+1})$ |

In Table 3, $\mathcal{L}_{x\theta}(x_{k+1}) := \frac{\partial}{\partial \theta} \nabla_x \mathcal{L}(x_{k+1}, s_k, \lambda_k, \nu_k; \theta)$, $P \succeq 0$ in the Quadratic layer and $H(x) = \sum_{i=1}^{n} x_i \log(x_i)$ is the negative entropy in the constrained Softmax layer. We next give the detailed derivation for the last column of Table 3.

*Proof.* For the Quadratic layer and the contrained Sparsemax layer, we substitute $\nabla f(x_{k+1}) = P x_{k+1} + q$ into (14) and calculate the derivative with respect to $\theta$, i.e.

$$P\frac{\partial x_{k+1}}{\partial \theta} + \frac{\partial q}{\partial \theta} + \frac{\partial A^T \lambda_k}{\partial \theta} + \frac{\partial G^T \nu_k}{\partial \theta} + \rho \frac{\partial A^T (A x_{k+1} - b)}{\partial \theta} + \rho \frac{\partial G^T (G x_{k+1} + s_k - h)}{\partial \theta} = 0. \tag{17}$$

Therefore we can derive the formulation of primal gradient update:

$$\begin{aligned}
\frac{\partial x_{k+1}}{\partial \theta} &= -\left(P + \rho A^T A + \rho G^T G\right)^{-1} \left(\frac{\partial q}{\partial \theta} + \frac{\partial A^T \lambda_k}{\partial \theta} + \frac{\partial G^T \nu_k}{\partial \theta} - \rho \frac{\partial (A^T b - G^T(s_k - h))}{\partial \theta}\right) \\
&= -\left(P + \rho A^T A + \rho G^T G\right)^{-1} \mathcal{L}_{x\theta}(x_{k+1}).
\end{aligned} \tag{18}$$

Compared with (16), we can find that the forward pass and backward pass in quadratic layers shared the same Hessian matrix, which significantly reduces the computational complexity.

The constrained Sparsemax Layer is a special quadratic layer that with sparse constraints, where $P = 2I$, $A = \mathbf{1}^T$ and $G = [-I, I]^T$. Therefore the primal differentiation (7a) becomes

$$\frac{\partial x_{k+1}}{\partial \theta} = -\left((2 + 2\rho)I + \rho \mathbf{1} \cdot \mathbf{1}^T\right)^{-1} \mathcal{L}_{x\theta}(x_{k+1}). \tag{19}$$

For constrained Softmax layer, the augmented Lagrange function is:

$$\begin{aligned}
\mathcal{L} = -y^T x + \sum_{i=1}^{n} x_i \log(x_i) + \lambda^T (\mathbf{1}^T x - 1) + \nu(Gx + s - h) \\
+ \frac{\rho}{2}\left(\|\mathbf{1}^T x - 1\|^2 + \|Gx + s - h\|^2\right),
\end{aligned} \tag{20}$$

where $G = [-I, I]^T$ and $h = [\mathbf{0}, \mathbf{u}]^T$. Taking the gradient of the first-order optimality condition (14), (17) can be replaced by

$$\mathbf{diag}^{-1}(x)\frac{\partial x_{k+1}}{\partial \theta} + \frac{\partial \mathbf{1}^T \lambda_k}{\partial \theta} + \frac{\partial G^T \nu_k}{\partial \theta} + \rho \frac{\partial \mathbf{1}(\mathbf{1}^T x_{k+1} - 1)}{\partial \theta} + \rho \frac{\partial G^T (G x_{k+1} + s_k - h)}{\partial \theta} = 0. \tag{21}$$

Similarly, the formulation of primal gradient update of the constrained Softmax layer is:

$$
\frac{\partial x_{k+1}}{\partial \theta} = - \left( \mathbf{diag}^{-1}(x) + \rho G^T G + \rho \mathbf{1} \cdot \mathbf{1}^T \right)^{-1} \left( \frac{\partial \mathbf{1}^T \lambda_k}{\partial \theta} + \frac{\partial G^T \nu_k}{\partial \theta} + \rho \frac{\partial G^T (s_k - h)}{\partial \theta} \right)
$$

$$
= - \left( \mathbf{diag}^{-1}(x) + 2\rho I + \rho \mathbf{1} \cdot \mathbf{1}^T \right)^{-1} \mathcal{L}_{x\theta}(x_{k+1}).
$$

(22)

$\square$

From Table 3, we can see the implementation of primal differentiation for the constrained Sparsemax layers Malaviya et al. (2018) and the quadratic optimization layers Amos & Kolter (2017) are quite simple. The Jacobian matrix involved in the primal update keeps the same during the iteration.

## C  ERROR OF THE TRUNCATED ALT-DIFF

**Theorem 4.1.** *(Error of gradient obtained by truncated Alt-Diff) Suppose $x_k$ is the truncated solution at the $k$-th iteration, the error between the gradient $\frac{\partial x_k}{\partial \theta}$ obtained by the truncated Alt-Diff and the real gradient $\frac{\partial x^\star}{\partial \theta}$ is bounded as follows,*

$$
\left\| \frac{\partial x_k}{\partial \theta} - \frac{\partial x^\star}{\partial \theta} \right\| \leq C_1 \|x_k - x^\star\|,
$$

(23)

*where $C_1 = \dfrac{L_3}{\mu_2} + \dfrac{\mu_3 L_2}{\mu_2^2}$ is a constant.*

*Proof.* In the differentiation procedure of Alt-Diff, we have our primal update procedure (7a), so that

$$
\frac{\partial x_k}{\partial \theta} - \frac{\partial x^\star}{\partial \theta} = -\nabla_x^2 \mathcal{L}^{-1}(x_k) \nabla_{x,\theta} \mathcal{L}(x_k) + \nabla_x^2 \mathcal{L}^{-1}(x^\star) \nabla_{x,\theta} \mathcal{L}(x^\star)
$$

$$
= -\nabla_x^2 \mathcal{L}^{-1}(x_k) \nabla_{x,\theta} \mathcal{L}(x_k) + \nabla_x^2 \mathcal{L}^{-1}(x_k) \nabla_{x,\theta} \mathcal{L}(x^\star)
$$

$$
- \nabla_x^2 \mathcal{L}^{-1}(x_k) \nabla_{x,\theta} \mathcal{L}(x^\star) + \nabla_x^2 \mathcal{L}^{-1}(x^\star) \nabla_{x,\theta} \mathcal{L}(x^\star)
$$

$$
= \nabla_x^2 \mathcal{L}^{-1}(x_k) \underbrace{\left( \nabla_{x,\theta} \mathcal{L}(x^\star) - \nabla_{x,\theta} \mathcal{L}(x_k) \right)}_{\Delta_1}
$$

(24)

$$
+ \underbrace{\left( \nabla_x^2 \mathcal{L}^{-1}(x^\star) - \nabla_x^2 \mathcal{L}^{-1}(x_k) \right)}_{\Delta_2} \nabla_{x,\theta} \mathcal{L}(x^\star).
$$

By Assumption (8c), we can obtain that

$$
\|\Delta_1\| = \|\nabla \mathcal{L}_{x,\theta}(x^\star) - \nabla \mathcal{L}_{x,\theta}(x_k)\| \leq L_3 \|x^\star - x_k\|.
$$

(25)

Since we have computed the closed form of the Hessian of augmented Lagrange function $\nabla_x^2 \mathcal{L}(x_k) = \nabla_x^2 f(x_k)^T + \rho A^T A + \rho G^T G$, by assumption (9) and Cauchy-Schwartz inequalities, we have

$$
\|\Delta_2\| = \|\nabla_x^2 \mathcal{L}^{-1}(x^\star) - \nabla_x^2 \mathcal{L}^{-1}(x_k)\|
$$

$$
= \|\nabla_x^2 \mathcal{L}^{-1}(x_k) \left( \nabla_x^2 \mathcal{L}(x_k) - \nabla_x^2 \mathcal{L}(x^\star) \right) \nabla_x^2 \mathcal{L}^{-1}(x^\star)\|
$$

$$
\leq \|\nabla_x^2 \mathcal{L}^{-1}(x_k)\| \cdot \|\nabla_x^2 \mathcal{L}(x_k) - \nabla_x^2 \mathcal{L}(x^\star)\| \cdot \|\nabla_x^2 \mathcal{L}^{-1}(x^\star)\|
$$

(26)

$$
\leq \frac{1}{\mu_2^2} \|\nabla_x^2 f(x^\star) - \nabla_x^2 f(x_k)\|.
$$

Therefore the difference of Jacobians is

$$
\left\| \frac{\partial x_k}{\partial \theta} - \frac{\partial x^\star}{\partial \theta} \right\| = \|\nabla_x^2 \mathcal{L}^{-1}(x_k) \cdot \Delta_1 + \Delta_2 \cdot \nabla_{x,\theta} \mathcal{L}(x^\star)\|
$$

$$
\leq \|\nabla_x^2 \mathcal{L}^{-1}(x_k)\| \cdot \|\Delta_1\| + \frac{1}{\mu_2^2} \|\nabla_{x,\theta} \mathcal{L}(x)\| \cdot \|\nabla_x^2 f(x^\star) - \nabla_x^2 f(x_k)\|
$$

(27)

$$
\leq \left( \frac{L_3}{\mu_2} + \frac{\mu_3 L_2}{\mu_2^2} \right) \|x^\star - x_k\|.
$$

$\square$

# D  ERROR OF THE GRADIENT OF LOSS FUNCTION

**Corollary 4.2.** *(Error of the inexact gradient in terms of loss function) Followed by Theorem 4.1, the error of the gradient w.r.t. $\theta$ in loss function $\mathcal{R}$ is bounded as follows:*

$$\|\nabla\mathcal{R}(\theta; x_k) - \nabla\mathcal{R}(\theta; x^\star)\| \leq C_2\|x_k - x^\star\|, \tag{28}$$

*where $C_2 = L_1 + \dfrac{\mu_3 L_1 + \mu_1 L_3}{\mu_2} + \dfrac{\mu_1\mu_3 L_2}{\mu_2^2}$ is a constant.*

*Proof.* Firstly, optimization layer can be reformulated as the following bilevel optimization problems Ghadimi & Wang (2018):

$$
\begin{aligned}
\min_\theta \quad & \mathcal{R}(\theta; x^\star(\theta)) \\
s.t. \quad & x^\star(\theta) = \arg\min_{x \in \mathcal{C}(\theta)} f(x; \theta),
\end{aligned} \tag{29}
$$

where $\mathcal{R}$ denotes the loss function, therefore we can derive the following conclusions by Implicit Function Theorem. In *outer optimization problems*, we have

$$\nabla\mathcal{R}(\theta; \tilde{x}) = \nabla_\theta\mathcal{R}(\theta; \tilde{x}) + \nabla_x\mathcal{R}(\theta; \tilde{x})\frac{\partial\tilde{x}}{\partial\theta}. \tag{30}$$

Therefore,

$$
\begin{aligned}
\Delta &= \nabla\mathcal{R}(\theta; x_k) - \nabla\mathcal{R}(\theta; x^\star) \\
&= \underbrace{\left(\nabla_\theta\mathcal{R}(\theta; x_k) - \nabla_\theta\mathcal{R}(\theta; x^\star)\right)}_{\Delta_3} + \underbrace{\left(\nabla_x\mathcal{R}(\theta; x_k)\frac{\partial x_k}{\partial\theta} - \nabla_x\mathcal{R}(\theta; x^\star)\frac{\partial x^\star}{\partial\theta}\right)}_{\Delta_4}.
\end{aligned} \tag{31}
$$

The second term is:

$$
\begin{aligned}
\Delta_4 &= \nabla_x\mathcal{R}(\theta; x_k)\frac{\partial x_k}{\partial\theta} - \nabla_x\mathcal{R}(\theta; x^\star)\frac{\partial x^\star}{\partial\theta} \\
&= \nabla_x\mathcal{R}(\theta; x_k)\frac{\partial x_k}{\partial\theta} - \nabla_x\mathcal{R}(\theta; x_k)\frac{\partial x^\star}{\partial\theta} \\
&\quad + \nabla_x\mathcal{R}(\theta; x_k)\frac{\partial x^\star}{\partial\theta} - \nabla_x\mathcal{R}(\theta; x^\star)\frac{\partial x^\star}{\partial\theta} \\
&= \underbrace{\left(\nabla_x\mathcal{R}(\theta; x_k) - \nabla_x\mathcal{R}(\theta; x^\star)\right)}_{\Delta_5}\frac{\partial x^\star}{\partial\theta} + \nabla_x\mathcal{R}(\theta; x_k)\left(\frac{\partial x_k}{\partial\theta} - \frac{\partial x^\star}{\partial\theta}\right).
\end{aligned} \tag{32}
$$

Combining the results in (32), assumption (8a) and Theorem 4.1, we can obtain that the differences of truncated gradient with optimal gradients is

$$
\begin{aligned}
\|\Delta\| &= \|\nabla\mathcal{R}(\theta; x_k) - \nabla\mathcal{R}(\theta; x^\star)\| \\
&\leq \|\Delta_3\| + \|\Delta_5\| \cdot \left\|\frac{\partial x^\star}{\partial\theta}\right\| + \|\nabla_x\mathcal{R}(\theta; x_k)\| \cdot \left\|\frac{\partial x_k}{\partial\theta} - \frac{\partial x^\star}{\partial\theta}\right\| \\
&\leq L_1\left(1 + \frac{\mu_3}{\mu_2}\right)\|x_k - x^\star\| + \mu_1\left(\frac{L_3}{\mu_2} + \frac{\mu_3 L_2}{\mu_2^2}\right)\|x_k - x^\star\| \\
&= \left(L_1 + \frac{\mu_3 L_1 + \mu_1 L_3}{\mu_2} + \frac{\mu_1\mu_3 L_2}{\mu_2^2}\right)\|x_k - x^\star\|.
\end{aligned} \tag{33}
$$

Therefore, the gradient of loss function $\mathcal{R}(\theta; x_k)$ shares the same order of error as the truncated solution $x_k$. $\square$

# E    ALT-DIFF AND DIFFERENTIATING KKT

We first consider the gradient obtained by differentiating KKT conditions implicitly, and then show that it is the same as the result obtained by Alt-Diff.

## E.1    DIFFERENTIATION OF KKT CONDITIONS

In convex optimization problem (1), the optimal solution is implicitly defined by the following KKT conditions:

$$
\begin{cases}
\nabla f(x) + A^T(\theta)\lambda + G^T(\theta)\nu = 0 \\
A(\theta)x - b(\theta) = 0 \\
\mathbf{diag}(\nu)(G(\theta)x - h(\theta)) = 0
\end{cases}
\tag{34}
$$

where $\lambda$ and $\nu$ are the dual variables, $\mathbf{diag}(\cdot)$ creates a diagonal matrix from a vector. Suppose $\tilde{x}$ collects the optimal solutions $[x^\star, \lambda^\star, \nu^\star]$, in order to obtain the derivative of the solution $\tilde{x}$ w.r.t. the parameters $A(\theta), b(\theta), G(\theta), h(\theta)$, we need to calculate $\mathcal{J}_{\mathcal{F};\tilde{x}}$ and $\mathcal{J}_{\mathcal{F};\theta}$ by differentiating KKT conditions (34) as follows Amos & Kolter (2017); Agrawal et al. (2019a); Zhang et al. (2020):

$$
\begin{bmatrix} \mathcal{J}_{\mathcal{F};x^\star} & | & \mathcal{J}_{\mathcal{F};\lambda} & | & \mathcal{J}_{\mathcal{F};\nu} \end{bmatrix} =
\begin{bmatrix}
f_{xx}^T(x^\star) & A^T(\theta) & B^T(\theta) \\
A(\theta) & 0 & 0 \\
\mathbf{diag}(\nu)G(\theta) & \mathbf{diag}(G(\theta)x - h(\theta)) & 0
\end{bmatrix}
\tag{35a}
$$

$$
\begin{bmatrix} \mathcal{J}_{\mathcal{F};A} & | & \mathcal{J}_{\mathcal{F};b} & | & \mathcal{J}_{\mathcal{F};G} & | & \mathcal{J}_{\mathcal{F};h} \end{bmatrix} =
\begin{bmatrix}
\nu^T \otimes I & 0 & \lambda^T \otimes I & 0 \\
I \otimes (x^\star)^T & -I & 0 & 0 \\
0 & 0 & \mathbf{diag}(\lambda)I \otimes (x^\star)^T & -I
\end{bmatrix}
\tag{35b}
$$

where $f_{xx}(x) := \nabla_x^2 f(x)$, $\otimes$ denotes the Kronecker product.

## E.2    ILLUSTRATION OF EQUIVALENCE

In this section, we show the equivalence of the Jacobian obtained by differentiating KKT conditions and by Alt-Diff from the computational perspective. As we have shown in Theorem 4.1, this section is a further complement of the equivalence. Here we show that the results obtained by Alt-Diff is equal to differentiating KKT conditions. Note the alternating procedure of (5) is based on ADMM, which is convergent for convex optimization problems. Assuming the optimal solutions by the alternating procedure of (5) are $[x_{\text{Alt}}^\star, s_{\text{Alt}}^\star, \lambda_{\text{Alt}}^\star, \nu_{\text{Alt}}^\star]$, we can derive the following results after the forward ADMM pass (5) converges:

$$
\begin{cases}
x_{\text{Alt}}^\star = \arg\min_x \mathcal{L}(x, s_{\text{Alt}}^\star, \lambda_{\text{Alt}}^\star, \nu_{\text{Alt}}^\star; \theta), \\
s_{\text{Alt}}^\star = \mathbf{ReLU}\left(-\frac{1}{\rho}\nu_{\text{Alt}}^\star - (Gx_{\text{Alt}}^\star - h)\right), \\
\lambda_{\text{Alt}}^\star = \lambda_{\text{Alt}}^\star + \rho(Ax_{\text{Alt}}^\star - b), \\
\nu_{\text{Alt}}^\star = \nu_{\text{Alt}}^\star + \rho(Gx_{\text{Alt}}^\star + s_{\text{Alt}}^\star - h).
\end{cases}
\tag{36}
$$

Taking the derivative with respect to the parameters $\theta$ in (36), the differentiation procedure (7) becomes

$$
\begin{cases}
\dfrac{\partial x_{\text{Alt}}^\star}{\partial \theta} = -\left(f_{xx}^T(x_{\text{Alt}}^\star) + \rho A^T A + \rho G^T G\right)^{-1} L_{x\theta}(x_{\text{Alt}}^\star), & (37a) \\[2mm]
\dfrac{\partial s_{\text{Alt}}^\star}{\partial \theta} = -\dfrac{1}{\rho}\mathbf{sgn}(s_{\text{Alt}}^\star) \cdot \mathbf{1}^T \odot \left(\dfrac{\partial \nu_{\text{Alt}}^\star}{\partial \theta} + \rho\dfrac{\partial(Gx_{\text{Alt}}^\star - h)}{\partial \theta}\right), & (37b) \\[2mm]
\dfrac{\partial \lambda_{\text{Alt}}^\star}{\partial \theta} = \dfrac{\partial \lambda_{\text{Alt}}^\star}{\partial \theta} + \rho\dfrac{\partial(Ax_{\text{Alt}}^\star - b)}{\partial \theta}, & (37c) \\[2mm]
\dfrac{\partial \nu_{\text{Alt}}^\star}{\partial \theta} = \dfrac{\partial \nu_{\text{Alt}}^\star}{\partial \theta} + \rho\dfrac{\partial(Gx_{\text{Alt}}^\star + s_{\text{Alt}}^\star - h)}{\partial \theta}. & (37d)
\end{cases}
$$

Recall that the differentiation through KKT conditions on the optimal solution $[x^\star, \lambda^\star, \nu^\star]$ are as follows,

$$
\begin{cases}
f_{xx}^T(x^\star)\dfrac{\partial x^\star}{\partial \theta} + \dfrac{\partial A^T \lambda^\star}{\partial \theta} + \dfrac{\partial G^T \nu^\star}{\partial \theta} = 0, & \text{(38a)} \\[4mm]
\dfrac{\partial (Ax^\star - b)}{\partial \theta} = 0, & \text{(38b)} \\[4mm]
\dfrac{\partial \nu^\star}{\partial \theta}(Gx^\star - h) + \nu^\star \dfrac{\partial (Gx^\star - h)}{\partial \theta} = 0. & \text{(38c)}
\end{cases}
$$

By the convergence of ADMM, it is known that $[x_{\mathrm{Alt}}^\star, \lambda_{\mathrm{Alt}}^\star, \nu_{\mathrm{Alt}}^\star]$ is equal to $[x^\star, \lambda^\star, \nu^\star]$. Since in convex optimization problems, the KKT condition (34) and its derivative (38) have unique solutions, therefore we only need to show that (38) can be directly derived by (37).

**(37a)$\Longrightarrow$(38a):**

$$
\begin{aligned}
& \left( f_{xx}^T(x_{\mathrm{Alt}}^\star) + \rho A^T A + \rho G^T G \right) \frac{\partial x_{\mathrm{Alt}}^\star}{\partial \theta} \\
= \quad & -\frac{\partial A^T \lambda_{\mathrm{Alt}}^\star}{\partial \theta} - \frac{\partial G^T \nu_{\mathrm{Alt}}^\star}{\partial \theta} + \rho \frac{\partial (A^T b - G^T (s_{\mathrm{Alt}}^\star - h))}{\partial \theta}.
\end{aligned}
\tag{39}
$$

Therefore,

$$
\begin{aligned}
& f_{xx}^T(x_{\mathrm{Alt}}^\star)\frac{\partial x_{\mathrm{Alt}}^\star}{\partial \theta} + \frac{\partial A^T \lambda_{\mathrm{Alt}}^\star}{\partial \theta} + \frac{\partial G^T \nu_{\mathrm{Alt}}^\star}{\partial \theta} \\
= \quad & -(\rho A^T A + \rho G^T G)\frac{\partial x_{\mathrm{Alt}}^\star}{\partial \theta} + \rho \frac{\partial (A^T b - G^T (s_{\mathrm{Alt}}^\star - h))}{\partial \theta} \\
= \quad & -\rho \frac{\partial (A^T (Ax_{\mathrm{Alt}}^\star - b))}{\partial \theta} - \rho \frac{\partial (G^T (Gx_{\mathrm{Alt}}^\star + s_{\mathrm{Alt}}^\star - h))}{\partial \theta} \\
= \quad & 0.
\end{aligned}
\tag{40}
$$

Clearly, (40) is exactly the same as (38a).

**(37c)$\Longrightarrow$(38b):** By (37c), we can obtain

$$
\frac{\partial (Ax_{\mathrm{Alt}}^\star - b)}{\partial \theta} = 0,
\tag{41}
$$

which is exactly the same as (38b).

**(37b) and (37d)$\Longrightarrow$(38c):** By (37b) and (37d), we can obtain:

$$
\begin{aligned}
\rho \frac{\partial (Gx_{\mathrm{Alt}}^\star - h)}{\partial \theta} &= -\rho \frac{\partial s_{\mathrm{Alt}}^\star}{\partial \theta} \\
&= \mathbf{sgn}(s_{\mathrm{Alt}}^\star) \cdot \mathbf{1}^T \odot \left( \frac{\partial \nu_{\mathrm{Alt}}^\star}{\partial \theta} + \rho \frac{\partial (Gx_{\mathrm{Alt}}^\star - h)}{\partial \theta} \right)
\end{aligned}
\tag{42}
$$

We discuss this issue in two situations. Since the equality $Gx_{\mathrm{Alt}}^\star + s_{\mathrm{Alt}}^\star - h = 0$ holds by optimality, we have $\langle \nu_{\mathrm{Alt}}^\star, s_{\mathrm{Alt}}^\star \rangle = 0$ using the complementary slackness condition.

- If $s_{\mathrm{Alt},i}^\star > 0$, then $\nu_{\mathrm{Alt},i}^\star = 0$, we obtain

$$
\begin{aligned}
\rho \frac{\partial (G_i x_{\mathrm{Alt}}^\star - h_i)}{\partial \theta} &= \mathbf{sgn}(s_{\mathrm{Alt},i}^\star) \cdot \mathbf{1}^T \odot \left( \frac{\partial \nu_{\mathrm{Alt},i}^\star}{\partial \theta} + \rho \frac{\partial (G_i x_{\mathrm{Alt}}^\star - h_i)}{\partial \theta} \right) \\
&= \mathbf{1} \cdot \mathbf{1}^T \odot \left( \frac{\partial \nu_{\mathrm{Alt},i}^\star}{\partial \theta} + \rho \frac{\partial (G_i x_{\mathrm{Alt}}^\star - h_i)}{\partial \theta} \right).
\end{aligned}
\tag{43}
$$

Therefore, $\dfrac{\partial \nu_{\mathrm{Alt},i}^\star}{\partial \theta} = 0$ and

$$
\frac{\partial \nu_{\mathrm{Alt},i}^\star}{\partial \theta}(G_i x_{\mathrm{Alt}}^\star - h_i) + \nu_{\mathrm{Alt},i}^\star \frac{\partial (G_i x_{\mathrm{Alt}}^\star - h_i)}{\partial \theta} = 0.
\tag{44}
$$

- If $s_{\text{Alt},i}^\star = 0$, then $G_i x_{\text{Alt}}^\star - h_i = 0$ and

$$\rho \frac{\partial (G_i x_{\text{Alt}}^\star - h_i)}{\partial \theta} = \text{sgn}(s_{\text{Alt},i}^\star) \cdot \mathbf{1}^T \odot \left( \frac{\partial \nu_{\text{Alt},i}^\star}{\partial \theta} + \rho \frac{\partial (G_i x_{\text{Alt}}^\star - h_i)}{\partial \theta} \right)$$

$$= \mathbf{0} \cdot \mathbf{1}^T \odot \left( \frac{\partial \nu_{\text{Alt},i}^\star}{\partial \theta} + \rho \frac{\partial (G_i x_{\text{Alt}}^\star - h_i)}{\partial \theta} \right) \qquad (45)$$

$$= 0.$$

Therefore,

$$\frac{\partial \nu_{\text{Alt},i}^\star}{\partial \theta} (G_i x_{\text{Alt}}^\star - h_i) + \nu_{\text{Alt},i}^\star \frac{\partial (G_i x_{\text{Alt}}^\star - h_i)}{\partial \theta} = 0. \qquad (46)$$

Therefore, (44) and (46) show (38c) holds.

## F EXPERIMENTAL DETAILS

In this section, we give more simulation results as well as some implementation details.

### F.1 ADDITIONAL NUMERICAL EXPERIMENTS

In constrained Sparsemax layer, the "lsqr" mode of CvxpyLayer is applied for fair comparison. The total running time of CvxpyLayer includes the canonicalization, forward and backward pass, retrieval and initialization. The comparison results are provided in Table 4. Alt-Diff can obtain the competitive results as OptNet and CvxpyLayer with lower executing time. The tolerances of the gradient are all set as $10^{-3}$ for OptNet, CvxpyLayer and Alt-Diff. Besides, we conduct experiments to compare the running time among Alt-Diff, OptNet and CvxpyLayer using different tolerances from $10^{-1}$ to $10^{-5}$ in OptNet, CvxpyLayer and Alt-Diff. The results in Table 5 shows that the truncated operation will not significantly improve the computational speed for existing methods but does offer significant improvements for Alt-Diff. All the numerical experiments were executed 5 times with the average times reported as the results.

Table 4: Comparison of running time (s) and cosine distances of gradients in constrained Sparsemax layers with tolerance $\epsilon = 10^{-3}$.

|  | Small | Medium | Large | Large+ |
|---|---|---|---|---|
| Num of variable $n$ | 3000 | 5000 | 10000 | 20000 |
| Num of constraints $n_c$ | 6000 | 10000 | 20000 | 40000 |
| **OptNet** | 18.46 | 77.87 | 834.59 | - |
| Forward | 16.61 | 71.45 | 779.00 | - |
| Backward | 1.85 | 6.42 | 55.59 | - |
| **CvxpyLayer (total)** | 3.61 | 18.14 | 123.79 | 1024.69 |
| Initialization | 3.50 | 17.79 | 122.61 | 1020.41 |
| Canonicalization | 0.00 | 0.00 | 0.01 | 0.02 |
| Forward | 0.07 | 0.27 | 0.44 | 1.35 |
| Backward | 0.03 | 0.07 | 0.73 | 2.91 |
| **Alt-Diff (total)** | **3.16** | **9.96** | **58.51** | **539.19** |
| Inversion | 1.21 | 4.71 | 34.35 | 287.83 |
| Forward and backward | 1.95 | 5.25 | 24.15 | 251.36 |
| Cos Dist. | 0.999 | 0.998 | 0.997 | 0.998 |

"-" represents that the solver cannot generate the gradients.

From Table 4, we can find that OptNet is much slower than CvxpyLayer in solving sparse Quadratic programmings. CvxpyLayer requires a lot of time in the initialization procedure, but once the initialization procedure completed, calling CvxpyLayer is very fast. However, Alt-Diff only needs to compute the inverse matrix once, significantly reducing the computational speed.

As a special case for general convex objective functions, we adopted $f(x) = -y^T x + \sum_{i=1}^n x_i \log(x_i)$, which is not a quadratic programming problem. Thus we can only compare Alt-Diff with CvxpyLayer. The constraints $Ax = b$ and $Gx \le h$ are randomly generated with dense

Table 5: Comparison of running time (s) for constrained Sparsemax layers with $n = 10000$ and $n_c = 20000$ under different tolerances.

| Tolerance $\epsilon$ | $10^{-1}$ | $10^{-2}$ | $10^{-3}$ | $10^{-4}$ | $10^{-5}$ |
|---|---|---|---|---|---|
| OptNet | 768.01 | 808.81 | 834.59 | 880.27 | 1026.85 |
| CvxpyLayer | 121.99 | 121.84 | 123.79 | 123.02 | 129.80 |
| Alt-Diff | 47.31 | 57.46 | 58.51 | 71.18 | 81.99 |

coefficients. In Alt-Diff, each primal update $x_{k+1}$ in (5a) is carried out by Newton's methods with the tolerance set as $10^{-4}$. The comparison of cosine distance and running time is in Table 6. We can find that Alt-Diff significantly outperforms in all these cases especially for optimization problems with large size.

Table 6: Comparison of running time (s) and cosine distances of gradients in constrained Softmax layers. Since OptNet only works for quadratic optimization problems, thus we cannot compare Alt-Diff with it for the constrained Softmax layers.

|  | tiny | small | medium | large |
|---|---|---|---|---|
| Num of ineq. $m$ | 100 | 300 | 500 | 1000 |
| Num of ineq. $m$ | 30 | 100 | 200 | 500 |
| Num of eq. $p$ | 10 | 20 | 20 | 200 |
| Elements | $1.96 \times 10^4$ | $1.76 \times 10^5$ | $5.18 \times 10^5$ | $2.89 \times 10^6$ |
| **CvxpyLayer, total** | 0.11 | 0.97 | 2.09 | 9.67 |
| Initialization | 0.01 | 0.04 | 0.08 | 0.29 |
| Canonicalization | 0.00 | 0.00 | 0.01 | 0.05 |
| Forward | 0.07 | 0.34 | 0.73 | 2.76 |
| Backward | 0.03 | 0.59 | 1.36 | 6.56 |
| **Alt-Diff, total** | 0.03 | 0.33 | 1.10 | 5.02 |
| Cosine Dist. | 0.999 | 0.996 | 0.999 | 0.999 |

"-" represents that the solver cannot generate the gradients.

## F.2 IMAGE CLASSIFICATION

All neural networks are implemented using PyTorch with Adam optimizer Kingma & Ba (2014). We replaced a layer as optimization layer in the neural networks, and compared the running time and test accuracy in MNIST dataset using OptNet and Alt-Diff. The batch size is set as $64$ and the learning rate is set as $10^{-3}$. We ran 30 epoches and provided the running time and test accuracy in Tabel 7. As we have shown that OptNet runs much faster than CvxpyLayer in dense quadratic layers, therefore we only compare Alt-Diff with OptNet.

Table 7: Comparison of OptNet and Alt-Diff on MNIST with tolerance $\epsilon = 10^{-3}$.

| models | Test accuracy (%) | Time per epoch (s) |
|---|---|---|
| OptNet | $98.95 \pm 0.25$ | $307.16 \pm 18.21$ |
| Alt-Diff | $98.97 \pm 0.17$ | $167.66 \pm 4.40$ |

We used two convolutional layers followed by ReLU activation functions and max pooling for feature extraction. After that, we added two fully connected layers with 200 and 10 neurons, both using the ReLU activation function. Among them, an optimization layer with an input of 200 dimensions and an output of 200 dimensions is added. We set the dimension of the inequality and equality constraints both as $50$. Following similar settings in OptNet, we used the quadratic objective function $f(x) = \frac{1}{2} x^T P x + q^T x$ here, while taking $q$ as the input of the optimization layer and the optimal $x^\star$ as the output. Finally, the 10 neurons obtained by the fully connected layer were input into Softmax layer, and the nonnegative log likelihood was used as the loss function here.

The experiment results are as follows, we can see Alt-Diff runs much faster and yields similar accuracy compared to OptNet.

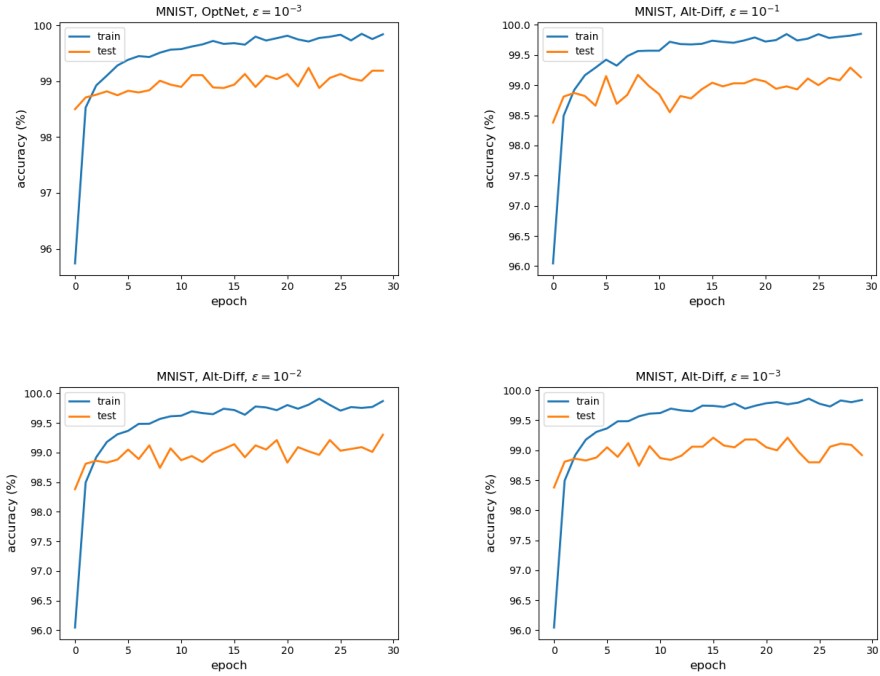

Figure 4: The training and testing performance of Alt-Diff and OptNet on MNIST.

Moreover, we show the training and testing performance of Alt-Diff and OptNet on MNIST in Figure 4. Obviously, Alt-Diff have similar performance under different tolerance values. Also, we can find that using a truncated threshold with low-accuracy in Alt-Diff will not sacrifice much accuracy, further validating the truncated capability of Alt-Diff as claimed in Corollary 4.2.

