# OpenReview forum: "Alternating Differentiation for Optimization Layers"
_ICLR.cc/2023/Conference — ICLR 2023 poster_

### Official Review · Reviewer_apdv · 2022-10-24

**Confidence:** 4
**Correctness:** 2
**Technical Novelty And Significance:** 3
**Empirical Novelty And Significance:** 3
**Recommendation:** 6

**Clarity, Quality, Novelty And Reproducibility:**

### Clarity
- The clarity of the paper is good, the ideas are clearly presented.
### Quality
- There are some weaknesses in quality as reported in strengths and weaknesses.
### Novelty
- While using ADMM in an algorithm for implicit differentiation is not novel in itself (e.g. it was used for solving the homogenous self-dual embedding of a cone program in [2], which is used in Cvxpy), the idea of implicitly differentiating the subproblems of ADMM in an iterative fashion to compute the jacobian in parallel to the solution of the optimization problem is novel to the best of my knowledge.
### Reproducibility
- The authors provide code for one of the experiments (sparsemax layer). I tested the code and was able to reproduce the results reported in Table 4. The authors promise to release the code for the remaining experiments upon acceptance.

[2]: Akshay Agrawal, Shane Barratt, Stephen Boyd, Enzo Busseti, and Walaa M Moursi. Differentiating through a cone program. arXiv preprint arXiv:1904.09043, 2019b.

**Strength And Weaknesses:**

### Strengths:
- The idea of using ADMM instead of competing methods to solve optimization problems in implicit layers is appealing, due to the simplicity, gneral applicability, and high parallelizability of ADMM.
- The paper is overall well written and well structured.
- The results are mostly convincing, with runtimes that are highly competitive with prior methods.

### Weaknesses:
While this paper proposes a method that brings clear advantages of ADMM to implicit layers, I believe that the claims of Alt-Diff being strictly better than previous methods are overstated.
(Examples: "Alt-Diff substantially decreases the dimensions of the Jacobian matrix and thus significantly increases the computational speed of implicit differentiation",  "comprehensive experiments demonstrate that Alt-Diff yields results comparable to the state-of-the-arts in far less time")

Firstly, the complexity analysis in Table 1 shows that in both prior methods and in Alt-Diff, the computation required for the backward pass is of the same order (or less) as the forward pass.  (Note that OptNet [1] even claims: "The backward pass gradients can be computed “for free” after solving the original QP with this primal-dual interior point method, without an additional matrix factorization or solve.")

Therefore the backward pass computation is not necessarily the computational bottleneck. And for the forward pass, from the complexities in Table 1 it is by no means obvious that using ADMM (Alt-Diff) is always better than using e.g. an Interior Point Method (OptNet), as there is a strong dependence of the practical runtimes on the number of iterations M and T needed for convergence.

This is also evident in the experiments, as the results for the sparsemax layer in Table 4 show that Cvxpy is much faster than Alt-Diff when disregarding the initialization, which to my understanding only needs to be performed once in the training procedure, so Cvxpy has a clear advantage over Alt-Diff in this setup.

In my opinion the story of the paper should be adjusted to accurately reflect that Alt-Diff does not make previous methods obsolete, and instead is one promising option among others, and choosing the best one highly depends on the problem at hand.

[1]: Brandon Amos and J Zico Kolter. Optnet: Differentiable optimization as a layer in neural networks. In International Conference on Machine Learning, pp. 136–145. PMLR, 2017.

### Additional questions:
- While I also believe that the idea of trading accuracy for runtime by truncation is very interesting, I do not think that this is inherent to ADMM and therefore could also be applied to speed up the competing methods. Have the authors tested increasing the tolerance to see if a similar speedup can be achieved?
- It would be useful to split the reported computation time into forward and backward pass also for OptNet.
- The results for OptNet are computed on a CPU, however, to my understanding OptNet was optimized for running on a GPU. This could potentially affect the runtime results. Have the authors attempted to run the experiments on a GPU?


### Minor issues:
- Equation 18, 19 (and text before): The computation misses the dependence on the parameter $q$. However, the final result in Equation 19 appears to be correct. A similar issue is in Equation 22, 23 with a missing dependence on $y$.
- Equation 32: $A$ and $G$ transposed missing in third line.
- Equation 35: First $\rho$ should be removed from second line.
- Equation 37: "opt" subscript is missing/missused.


**Summary Of The Paper:**

This paper proposes Alternating Differentiation for Optimization Layers (Alt-diff), a method for differentiation of parameterized optimization problems. Prior methods first solve the optimization problem on the forward pass using e.g. interior point methods, and then implicitly differentiate the KKT conditions on the backward pass by solving a linear system of equations.

Alt-diff is based on instead using ADMM to solve the optimization problem on the forward pass by decoupling it into multiple subproblems, which are then solved in an iterative alternating fashion. The jacobian can then be computed simultaneously to the forward pass, by differentiating the subproblems in the same iterative alternating fashion as the forward computation. The main benefit from this new approach is a computational speedup due to the reduced complexity of implicitly differentiating the subproblems with a smaller KKT matrix, compared to directly differentiating the original problem.

The authors show that the backward pass enjoys quadratic complexity. They also show that the runtime can further be improved by truncating the iterative procedure. This gives a tradeoff between the error on the jacobian and the runtime, which is also theoretically analysed in the paper.

The proposed method is experimentally tested on multiple experiments, including a runtime analysis of optimization layers with randomly generated parameters, an energy scheduling problem in predict-then-optimize fashion, and an image classification task. The experiments show that the proposed method can often yield results of quality similar to state-of-the-art methods in less time.

**Summary Of The Review:**

The idea of reducing the computational complexity of implicit differentiation by decomposing the optimization problem in an ADMM style is exciting.
My main concern is that the authors overstate the general superiority of the proposed method and do not highlight enough the importance of choosing an appropriate algorithm for the optimization problem at hand.
If the remaining issues regarding the presentation of the comparison to previous works are resolved, I believe this paper will be of great interest to the ICLR community.

---

> ### Author Response · Authors · 2022-11-18
> **Reply to reviewer apdv (Part 2 of 2)**
>
> > Q5: In my opinion the story of the paper should be adjusted to accurately reflect that Alt-Diff does not make previous methods obsolete, and instead is one promising option among others, and choosing the best one highly depends on the problem at hand.
>
> A5: Thanks for your constructive comments. We agree that Alt-Diff is an alternative choice for differentiating convex optimization problems. The main advantage of Alt-Diff comes from its capability to efficiently handle optimization problems with a large number of constraints. Therefore, we believe Alt-Diff will not make existing methods obsolete. In addition, Alt-Diff enjoys good truncated capability, i.e., we can truncate Alt-Diff to further accelerate the computational speed without sacrificing much accuracy. We have modified the Abstract, Contributions and Conclusion accordingly.
> > Q6: While I also believe that the idea of trading accuracy for runtime by truncation is very interesting, I do not think that this is inherent to ADMM and therefore could also be applied to speed up the competing methods. Have the authors tested increasing the tolerance to see if a similar speedup can be achieved?
>
> A6: Thanks for your comments. The truncation for existing methods (OptNet and CvxpyLayers) that are based on the interior point methods often does not have significant improvements. This is because the interior point methods often converge fast whereas the computational bottleneck comes from the high dimension of the KKT matrix instead of the number of iterations of the interior points methods. However, the iteration truncation can benefit Alt-Diff more as the iteration of Alt-Diff is more related to the tolerance due to the sublinear convergence of ADMM.
>
> We also conduct experiments on a sparsemax layer and a dense quadratic layer respectively to show this phenomenon and present the simulation results in the following tables.
>
>
> **Sparsemax, n=10000**
> | Tolerance $\epsilon$	|1e-1   |1e-2    |1e-3  | 1e-4	|1e-5|
> | :----:   | :----: | :----: | :----:| :----: | :----: |
> | OptNet   |768.01 	|808.81  |834.59 |880.27 |1026.85 |
> |CvxpyLayer|121.99 	|121.84 	|123.79 	|123.02 	|129.80 |
> |Alt-Diff	|47.31 	|57.46 	|58.51 	|71.18 |	81.99 |
>
> **Dense Quadratic, n=3000, m=1000, p=500**
>
> | Tolerance $\epsilon$	|1e-1   |1e-2    |1e-3  | 1e-4	|1e-5|
> | :----:   | :----: | :----: | :----:| :----: | :----: |
> |OptNet	|8.58 	|9.32 	|9.48 	|9.59 	|10.13|
> |CvxpyLayer	|212.34 	|216.66 	|212.86 	|215.75 	|213.08|
> |Alt-Diff	|1.18 	|3.50 	|5.75 	|8.04 	|10.76|
>
> From the tables, we can find that the truncated operations will not significantly improve the computational speed for existing methods but do have significant improvements for Alt-Diff. The above analysis and simulation results have been added in Section 5.1 and Appendix F.1 of the revision.
>
>
> > Q7: It would be useful to split the reported computation time into forward and backward pass also for OptNet.
>
> A7: We have re-run the code of OptNet and added the running time of forward and backward pass for OptNet in Table 1 and Table 4.
>
> > Q8: The results for OptNet are computed on a CPU, however, to my understanding OptNet was optimized for running on a GPU. This could potentially affect the runtime results. Have the authors attempted to run the experiments on a GPU?
>
> A8: Yes, we are trying to implement Alt-Diff on a GPU but have not finished yet. We will release the code and shall add a GPU version of Alt-Diff soon.
>
> > Q9: About typos.
>
> A9: Thanks for your careful reading! We appreciate the minor suggestions and have improved the paper accordingly.

---

> > ### Comment · Reviewer_apdv · 2022-12-07
> > **Response to Rebuttal**
> >
> > I thank the authors for the detailed response to my review, which has addressed all the concerns that I have raised. I am happy with the updates made to the revised version, and taking into account the other reviews and author responses, I also vote for acceptance.

---

> ### Author Response · Authors · 2022-11-18
> **Reply to reviewer apdv (Part 1 of 2)**
>
> We would like to thank the reviewer for the time and expertise invested in these reviews, and we really appreciate your constructive comments. Here are our responses to the reviewer's concerns.
>
> > Q1: While this paper proposes a method that brings clear advantages of ADMM to implicit layers, I believe that the claims of Alt-Diff being strictly better than previous methods are overstated. (Examples: "Alt-Diff substantially decreases the dimensions of the Jacobian matrix and thus significantly increases the computational speed of implicit differentiation", "comprehensive experiments demonstrate that Alt-Diff yields results comparable to the state-of-the-arts in far less time")
>
> A1: In the revision, we have emphasized the superiority of Alt-Diff, especially on the large-scale optimization problem. The abstract and contributions have been revised accordingly. The fast performance of Alt-Diff comes from the following two aspects: 1) Alt-Diff reduces the dimension of the KKT matrix by decoupling the objective functions and constraints, therefore it is more suitable to large-scale problems; 2) the truncated capability: the truncation error of gradients is upper bounded by the same order of variables' given some standard assumptions, therefore, Alt-Diff can be truncated to accelerate the computational speed without scarifying much accuracy.
>
> > Q2: The complexity analysis in Table 1 shows that in both prior methods and in Alt-Diff, the computation required for the backward pass is of the same order (or less) as the forward pass. (Note that OptNet [1] even claims: "The backward pass gradients can be computed “for free” after solving the original QP with this primal-dual interior point method, without an additional matrix factorization or solve.")
>
> A2: We agree the backward pass of OptNet is much easier than the forward pass due to the application of the primal-dual interior point method in the forward pass.  The computational complexity for the backward pass is $\mathcal{O}\big((n+n_c)^2\big)$, which is indeed quite smaller compared to the forward complexity as $\mathcal{O}\big(T(n+n_c)^3\big)$. In the revision, we have highlighted that “the total complexity (forward & backward) for Alt-Diff is $\mathcal{O}\big(n^3\big)$, which differs from directly differentiating KKT conditions with complexity as $\mathcal{O}\big((n+n_c)^3\big)$”. The original Table I has been deleted to avoid misleading.
>
>
> > Q3: Therefore the backward pass computation is not necessarily the computational bottleneck.
> And for the forward pass, from the complexities in Table 1 it is by no means obvious that using ADMM (Alt-Diff) is always better than using e.g. an Interior Point Method (OptNet), as there is a strong dependence of the practical runtimes on the number of iterations M and T needed for convergence.
>
> A3: We agree with you that the backward pass for OptNet is easy. However, this is not the case for more general optimization problems as OptNet only works for quadratic optimization problems. The computation in the backward pass of CvxpyLayer is nontrivial. Although CvxpyLayer applied LSQR to accelerate the backward pass, this method is only limited to optimization problems with specific sparse structures. Therefore, we believe Alt-Diff still has its place in efficiently processing the backward pass. With regards to the forward pass, generally, the iteration required by Alt-Diff is larger than the prior methods which are based on the interior point methods, i.e., $M>T$. As shown in [Section 3.2.2, Boyd et al 2011], ADMM often converges within a few tens of iterations to modest accuracy. Therefore, Alt-Diff has a clear advantage for moderate and large optimization problems as $n_c \gg M$.
>
> > Q4: This is also evident in the experiments, as the results for the sparsemax layer in Table 4 show that CvxpyLayer is much faster than Alt-Diff when disregarding the initialization, which to my understanding only needs to be performed once in the training procedure, so CvxpyLayer has a clear advantage over Alt-Diff in this setup.
>
> A4: CvxpyLayer uses a warm start mechanism to accelerate the solving procedure by reusing the cached KKT matrix factorization. Note that the weights of neural networks are updated frequently in the training procedure. Hence the initialization procedure needs to be re-implemented frequently as the KKT matrix changes after the update of neural networks. In addition, the initialization acceleration is not quite useful when the optimization problem is not sparse. As shown in Table 1 (Table 2 of the original version), even ignoring the initialization time, the running time of CvxpyLayer is still much larger than Alt-Diff for dense quadratic layers.

---

### Official Review · Reviewer_kA5n · 2022-10-24

**Confidence:** 4
**Correctness:** 2
**Technical Novelty And Significance:** 4
**Empirical Novelty And Significance:** 2
**Recommendation:** 6

**Clarity, Quality, Novelty And Reproducibility:**

To my knowledge, using ADMM in optimization layers on both forward and backward passes is a novel and significant contribution. The idea is clearly presented and easy to follow. I have some concerns about the contribution exaggeration and quality of the technical part; see the weaknesses above. I have not run the supplemented code alone, but I do not expect any surprises in reproducing the reported numbers.

**Strength And Weaknesses:**

Strengths:
- Approaches an important and difficult problem with a different viewpoint by using ADMM
- Convincing evaluation showing performance improvement over SOTA in some cases
- Well-written related work providing enough background to understand the paper

Weaknesses:
- As I understood the paper's message, it claims the supremacy of Alt-Diff over OptNet and cvxpylayer in all cases. However, this is overstated:
	- OptNet and cvxpylayers never compute the full Jacobian matrix on backward (as claimed in the abstract and also later in section 3). In fact, from how the forward is computed, the backward is "for free" (see the first line of 3.1.1. of OptNet paper), i.e. O(1). This is also true for Alt-Diff; there is, in fact, no backward pass. In any case, only the larger of the two (forward/backward) counts.
	- Hence, we are comparing $O(T(n+n_c)^3)$ of cvxpylayer (interior point method) to $O(Mn^3)$ of Alt-Diff (ADMM). What ultimately decides is the number of iterations $T$ and $M$. Therefore, one should use the method which is suitable for a given optimization problem: If the ADMM converges faster, Alt-Diff is the right method, and if the interior point method works better, cvxpylayer should be used.
	- 'Additional numerical experiments' in Appendix F.1 on sparse quadratic programs actually proves my point: cvxpylayer is 30x faster than Alt-diff since the initialization needs called only once.
- Convergence analysis. It is evident from the experiments (Fig. 2a) that the gradient is correct. Unfortunately, I do not see how it follows from the proof of Theorem 4.2.
	- I do not understand where Lemma 4.1 is used. Also, why from all possible fixed point theorems, this particular one is quoted (for reflexive Banach spaces). The assumption should also be 1-Lipschitz (nonexpansive) mapping from a bounded closed convex set into itself.
	- Thm 4.2: The proof contains computations showing that differentiating optimality conditions of ADMM leads to the same analytical derivative as differentiating the KKT conditions (this seems to be more or less correct). Any proof that the convergence of values (guaranteed by the convergence of ADMM in the convex case) yields the convergence of derivatives is missing (referred as 'obvious'). In general, convergence (even uniform) of functions does not imply convergence of derivatives (see eg. https://en.wikipedia.org/wiki/Uniform_convergence#To_differentiability). It is not, in my opinion, obvious, and it should be clarified.
	- Theorems 4.3 and 4.4 present expected results under very strong assumptions with elementary estimates. However, in particular, estimate 11 proves the convergence of derivatives in this case.

sugesstions, typos etc.:
- eq. 6: is $s\ge 0$ correct here?
- Thm. 4.2: $\partial x^*/\partial\theta$ denotes the true analytical Jacobian; there is no need to refer to cvxpy.
- Algo. 1: The elements of the last iteration should be returned.
- Tab. 1: Is $k$ correct here? ($=M$?)
- Section 5.1 Matrix $P$ seems to be only defined later in the appendix.

**Summary Of The Paper:**

An optimization layer is a function that assigns to given parameters $\theta$ the solution $x^*$ of an optimization problem $\min_x f(x,\theta)$ subject to constraints $x\in C(\theta)$. These layers are also implicit layers as $x^*$ is a solution of a certain implicitly defined function $\cal F(x^*,\theta)=0$. Under reasonable assumptions, these layers have analytical gradients $\partial x^*/\partial\theta$, and they can be used in end-to-end trainable NN models. The aim is to study how to calculate the forward and backward passes of these layers efficiently.

The paper studies the alternating direction method of multipliers (ADMM) for convex optimization layers ($f$ is convex). On the forward pass, at every ADMM iteration, the primal and dual variables and their Jacobians are updated until convergence. The paper contains theoretical justification, complexity analysis and comparison to OptNet and Cvxpylayers (using the interior point method), and the performance is demonstrated and compared on a synthetic and two real-world datasets.

**Summary Of The Review:**

The paper persuasively shows that differentiating ADMM leads to better runtimes in optimization layers where ADMM is the appropriate algorithm. If the authors address the abovementioned issues, the paper will be a valuable asset to the community.

---

> ### Author Response · Authors · 2022-11-18
> **Reply to reviewer kA5n (Part 1 of 2)**
>
> We thank the reviewer for your valuable feedback and constructive comments! We itemize the weaknesses or comments you mentioned and answer to them.
>
> > Q1: As I understood the paper's message, it claims the supremacy of Alt-Diff over OptNet and cvxpylayer in all cases. However, this is overstated:
> OptNet and cvxpylayers never compute the full Jacobian matrix on backward (as claimed in the abstract and also later in section 3). In fact, from how the forward is computed, the backward is "for free" (see the first line of 3.1.1. of OptNet paper), i.e. O(1). This is also true for Alt-Diff; there is, in fact, no backward pass. In any case, only the larger of the two (forward/backward) counts.
>
> A1: Thanks for your comments! In the revision, we claimed the main supremacy of Alt-Diff comes from its advantage on the large-scale constraints and its truncated capability to further reduce the computational speed. Although the authors claimed the backward pass of OptNet can be computed “for free”, but unfortunately OptNet does not have O(1) complexity in the backward pass. The actual computational complexity for the backward pass is $\mathcal{O}\big((n+n_c)^2\big)$. Besides, we have added the detailed computational time of the backward pass of OptNet in Table 1 and Table 4.
>
> > Q2: Hence, we are comparing $\mathcal{O}\big(T(n+n_c)^3\big)$ of cvxpylayer (interior point method) to $\mathcal{O}\big(Mn^3\big)$ of Alt-Diff (ADMM). What ultimately decides is the number of iterations T and M. Therefore, one should use the method which is suitable for a given optimization problem: If the ADMM converges faster, Alt-Diff is the right method, and if the interior point method works better, cvxpylayer should be used.
>
> A2: Thank you for raising the concern. The iteration required by Alt-Diff is generally larger than the prior methods which are based on the interior point methods, i.e., $M>T$. However, as shown in [Section 3.2.2, Boyd et al 2011], ADMM often converges within a few tens of iterations to modest accuracy. Therefore, Alt-Diff has a clear advantage for moderate and large optimization problems as $n_c \gg M$.
>
> We agree the forward pass has an important effect on differentiating optimization problems. However, this does not mean this is the only criterion to choose suitable methods simply because the differentiation procedure in the backward pass is nontrivial for general optimization problems. One needs to consider both the forward pass and backward pass to select suitable methods.
>
>
> > Q3: 'Additional numerical experiments' in Appendix F.1 on sparse quadratic programs actually proves my point: cvxpylayer is 30x faster than Alt-diff since the initialization needs called only once.
>
> A3: CvxpyLayer uses a cached matrix factorization mechanism as a "warm start" to reduce the computational speed. However, the advantages of the initialization cannot be reflected in training neural networks, because the parameters $\theta$ will be updated frequently during the training procedure. In this case, the initialization procedure of CvxpyLayer needs to be frequently re-implemented. Besides, the initialization acceleration for dense cases is not quite helpful as shown in Table 1 (Table 2 of the original version). Even ignoring the initialization time, the running time of CvxpyLayer is still much larger than Alt-Diff for dense quadratic layers.
>
>
> > Q4: Convergence analysis. It is evident from the experiments (Fig. 2a) that the gradient is correct. Unfortunately, I do not see how it follows from the proof of Theorem 4.2.
> I do not understand where Lemma 4.1 is used. Also, why from all possible fixed point theorems, this particular one is quoted (for reflexive Banach spaces). The assumption should also be 1-Lipschitz (nonexpansive) mapping from a bounded closed convex set into itself.
>
> A4: Thanks for your advice! The Lemma 4.1 is redundant and has been deleted in our revision. Moreover, we also revised the expression in Theorem 4.2 (Theorem 4.1 in the revision). The original expression in Theorem 4.2 was not exact, the actual purpose of Theorem 4.2 is to show the obtained gradients by Alt-Diff are consistent with the ones obtained by differentiating KKT.

---

> > ### Comment · Reviewer_kA5n · 2022-11-30
> > **Reply**
> >
> > Thank you for your detailed explanation, clarification and helpful updates to the paper. I am convinced that the paper brings a valuable contribution to the discussion of optimization layers, and I would like to see it accepted.
> >
> > There is just one minor thing that I would like to discuss and clarify with the authors, and that is the statement of Theorem 4.1. I want to stress that, in my opinion, this theorem it is not crucial for the paper and should not play any decisive role in the acceptance.
> > - Function $\theta\mapsto x^*(\theta)$ is differentiable, and $\partial x^*/\partial\theta$ is its analytical Jacobian (or derivative, 'gradient' may be misleading and confusing with $\partial\mathcal R/\partial\theta)$. By $\partial x^*/\partial\theta$ should always be meant this unique object.
> > 	- This is related to my comment, "Algo. 1: The elements of the last iteration should be returned" since, as I understand it, Algo. 1 stops at the first $k$ for which $\|x_{k+1}-x_{k}\|<\epsilon\|x_k\|$ and $x_{k+1}$ and $\partial x_{k+1}/\partial\theta$ are returned instead of the true (unknown) $x^*$ and its Jacobian. In theory, these should be distinguished.
> > - Solving a system given by KKT conditions is only one of the methods to compute $\partial x^*/\partial\theta$, so '...obtained by differentiating KKT conditions' is redundant in the statement (also in Theorem 4.2).
> > - Now to the main thing:
> > > A5: ...we just prove that when the backward pass (5) reaches its final results, the obtained gradients are consistent with the derivative...
> > - I agree that you proved that when (5) stabilizes, its differentiation leads to a system equivalent to KKT conditions.
> > 	- Actually, this we can show without any computations: By the assumptions, ADMM converges and hence the solution $x_{\hbox{opt}}$ of (27) coincides with $x^*$ (they are the same objects). Consequently, $\partial x_{\hbox{opt}}/\partial\theta=\partial x^*/\partial\theta$.
> > 	- It also means that (27) implicitly defines the very same function as KKT conditions and, therefore, its Jacobian has to solve both systems simultaneously.
> > - However, the theorem says 'Let $\partial x_{\hbox{opt}}/\partial\theta$ denote the final results obtained by the primal differentiation (7a)':
> > 	- I is not clear to me what it exactly refers to. As I understand it, (7a) defines a sequence $\partial x_{k}/\partial\theta$ for $k\in\mathbb N$. Does then $\partial x_{\hbox{opt}}/\partial\theta$ denote its limit?
> > 	- If so, I do not see that it even exits (eg. divergent or oscilating sequences). It is mentioned that $x_k$ is convergent for convex problems. Is that also true for $\partial x_{k}/\partial\theta$? If so, it would deserve a reference.
> > 	- Next, even if it converges, the relation of the limit $\partial x_{\hbox{opt}}/\partial\theta$ to the Jacobian $\partial x^*/\partial\theta$ is again unclear to me. From (39), it seems that continuity of $\nabla_{x,\theta}\mathcal L$ and $\nabla^2_x\mathcal L^{-1}$ is sufficient (maybe also necessary?).
> > 	- I believe we both agree that interchanging limits and derivatives is not possible without assumptions (o:
> > - I suggest omitting this theorem and focusing on Theorem 4.2 which guarantees convergence. I believe that it is much more relevant and shows that Algo. 1 returns a reasonable approximation of the Jacobian.

---

> > > ### Author Response · Authors · 2022-12-02
> > > **Further reply**
> > >
> > > Thank you for the time and thorough reviews. We are really happy to see you like our methods and contributions. Here are our responses to your minor comments.
> > >
> > > > Algo. 1: The elements of the last iteration should be returned.
> > >
> > > We will revise the return of Algorithm 1 as you suggested.
> > >
> > > > Solving a system given by KKT conditions is only one of the methods to compute ${\partial x^\star}/{\partial \theta}$, so '...obtained by differentiating KKT conditions' is redundant in the statement (also in Theorem 4.2).
> > >
> > > We will delete such expression of 'differentiating KKT conditions' in the final version.
> > >
> > > > Does then ${\partial x_{opt}}/{\partial \theta}$ denote its limit?
> > >
> > > Yes, $x_{opt}$ means the optimal value obtained by ADMM procedure (5a) and $\frac{\partial x_{opt}}{\partial \theta}$ is its corresponding Jacobian derived by (7a). To avoid misleading, we will delete the symbol of $x_{opt}$ and only use $x^{\star}$ to represent the optimal value in the final version.
> > >
> > > > About the other concerns about Theorem 4.1.
> > >
> > > Theorem 4.1 shows that the Jacobian $\partial x_{\hbox{opt}}/\partial\theta=\partial x^\star/\partial\theta$. We agree with you that a proof that $\lim\limits_{k\rightarrow \infty }\frac{\partial x_{k}}{\partial \theta}=\frac{\partial x_{opt}}{\partial \theta}$ is an important premise for illustrating the equivalence of the two Jacobians. Actually, we did have such proof in Theorem 4.2.
> > >
> > > Theorem 4.2 shows that $\|\frac{\partial x_k}{\partial \theta}-\frac{\partial x^\star}{\partial \theta}\|\leq C \|x_k-x^\star\|$ for constant $C$ under some standard assumptions (Please also refer the detailed derivations in Eq. 42). As you expected, the assumption includes the Lipschitz continuous of the $\nabla_{x\theta}\mathcal{L}(x)$ and $\nabla^2_x \mathcal{L}(x)$, which are the sufficient condition for the convergence of Jacobians. By the convergence of ADMM itself, $x_k$ will converge to $x^\star$, i.e., $\forall \epsilon>0$, $\exists k$, s.t. $\|x_k-x^\star \|<\epsilon$, therefore $\|\frac{\partial x_k}{\partial \theta}-\frac{\partial x^\star}{\partial \theta}\|\leq C\epsilon$. Accordingly, the convergence of the Jacobian is established.
> > >
> > > Given the convergence of the Jacobian by Theorem 4.2, Theorem 4.1 shows the equivalence of the Jacobian obtained by differentiating KKT conditions and by Alt-Diff from the computational perspective. We agree with you this may be redundant. In the final version, we will focus on Theorem 4.2 in the main contexts while modifying Theorem 4.1 to a remark and leaving it in the Appendix.

---

> > > > ### Comment · Reviewer_kA5n · 2022-12-02
> > > > **Reply to further reply**
> > > >
> > > > Thank you for your quick reaction.  I believe that these (minor) changes will make the paper even nicer and I vote for acceptance.

---

> ### Author Response · Authors · 2022-11-18
> **Reply to reviewer kA5n (Part 2 of 2)**
>
> > Q5: Thm 4.2: The proof contains computations showing that differentiating optimality conditions of ADMM leads to the same analytical derivative as differentiating the KKT conditions (this seems to be more or less correct). Any proof that the convergence of values (guaranteed by the convergence of ADMM in the convex case) yields the convergence of derivatives is missing (referred as 'obvious'). In general, convergence (even uniform) of functions does not imply convergence of derivatives (see eg. https://en.wikipedia.org/wiki/Uniform_convergence#To_differentiability). It is not, in my opinion, obvious, and it should be clarified.
>
> A5: Thanks for your constructive comments. We would like to clarify that the purpose of this Theorem 4.2 (Theorem 4.1 in the revision) is not to show the convergence of differentiation. Instead, we just prove that when the backward pass (5) reaches its final results, the obtained gradients are consistent with the derivative by differentiating KKT, i.e. $\frac{\partial x_\text{opt}}{\partial \theta}$ = $\frac{\partial x^\star}{\partial \theta}$. In the revision, we have revised the expression of this theorem as well as the related proof in the appendix.
>
> > Q6: Theorems 4.3 and 4.4 present expected results under very strong assumptions with elementary estimates. However, in particular, estimate 11 proves the convergence of derivatives in this case.
>
> A6: The current assumptions are strong but quite standard for convergence analysis. The estimate 11 can show the convergence of derivatives based on these assumptions.
>
> > Q7: About typos
>
> A7: Thanks for your valuable comments! We have modified our draft according to your advice.
> 1. $s\geq0$: We have deleted $s\geq0$ in the Relu Operator.
> 2. We have deleted the reference to CvxpyLayer.
> 3. Alg.1: We do not quite understand this comment as we already “Return $x^\star$ and its gradient $\frac{\partial x^\star}{\partial \theta}$” in Alg.1 .
> 4. We have deleted Table 1 and explained the reasons in the Reply A2 for reviewer apdv.
> 5. We have defined the quadratic objective functions in Section 5.1.

---

### Official Review · Reviewer_mj5g · 2022-11-02

**Confidence:** 4
**Correctness:** 4
**Technical Novelty And Significance:** 3
**Empirical Novelty And Significance:** 3
**Recommendation:** 8

**Clarity, Quality, Novelty And Reproducibility:**

From my point of view, this paper is well-written and comprehensible. The authors provide an appropriate introduction and review of related work so that the paper is more or less self-contained. As far as I can see the results are novel, and the authors have published their source code.

**Strength And Weaknesses:**

Strengths:

Compared to existing approaches that use implicit differentiation, namely OptNet and CvxpyLayer, the proposed framework needs no differentiation through KKT conditions which reduces the dimension of the required inverse Hessian by the number of constraints of the optimization problem. As a byproduct, the inverse Hessian for differentiation can be reused from the preceding ADMM update. This reduces the complexity of the backward pass from cubic to quadratic. Compared to unrolling, the proposed method does not require that intermediate iterates from the forward pass are stored for backpropagation as gradients are computed on the fly. The presented numerical results suggest that Alt-Diff yields state-of-the-art accuracy on benchmark problems while saving significant computational budget.

Weaknesses:

The proposed approach still needs inverse Hessians. Through the decoupling of objective and constraints, the scalability in terms of the number of constraints is improved. On the other hand, if I see it correctly, the scalability in terms of the number of optimization variables is not better than in case of existing approaches. Moreover, the threshold $\epsilon$ needed for the stopping criterion constitutes an additional tuning parameter which might not be obvious to choose in practice.

**Summary Of The Paper:**

The authors propose Alt-Diff, a framework for differentiation of optimization layers in neural networks. The basic idea is to write down the augmented Lagrangian for the considered class of convex optimization problems with polyhedral constraints, derive associated ADMM updates in which objective and constraints are decoupled, and therefrom gradients of primal, slack and dual variables w.r.t. optimization parameters. The resulting recursive sequence of gradients is theoretically shown to converge (in terms of the number of ADMM iterations) towards the gradient obtainened by implicit differentiation. In addition, the authors provide a complexity analysis of the proposed method, theoretical upper bounds on the error in gradient computation introduced by truncation of the recursive scheme after a finite number of iterations, and numerical experiments.

**Summary Of The Review:**

All in all, I think that this is a very good paper. The authors provide a tractable approach for the differentiation of optimization layers accompanied by a rigorous theoretical analysis, and the numerical results seem convincing to me. I would say that the above-mentioned weaknesses are rather mild, and the approach stands well for itself in comparison with existing approaches.

---

> ### Author Response · Authors · 2022-11-18
> **# Reply to reviewer mj5g**
>
> We really thank the reviewer for the insightful and positive comments!
>
> > The proposed approach still needs inverse Hessians. Through the decoupling of objective and constraints, the scalability in terms of the number of constraints is improved. On the other hand, if I see it correctly, the scalability in terms of the number of optimization variables is not better than in case of existing approaches.
>
> Thanks for your comments. The inverse Hessians are only used in the forward pass, which is hard to be avoided unless using some approximation methods. Alt-Diff has an obvious advantage when handling large-scale problems with many constraints and shares similar scalability with existing methods in terms of the number of optimization variables. Nonetheless, the truncated capability of Alt-Diff is another merit that can reduce its computational time without losing much accuracy. Note that the truncation for OptNet and CvxpyLayers is not quite significant in terms of accelerating the computation speed, as the bottleneck comes from the high dimension of the KKT matrix instead of the iteration numbers of the interior points methods.
>
> > Moreover, the threshold needed for the stopping criterion constitutes an additional tuning parameter which might not be obvious to choose in practice.
>
> Alt-Diff is not quite sensitive to the selection of stopping criterion. In general, we just choose a quite common stopping criterion, such as 1e-3 or 1e-4. We also conducted experiments to show the model performance under various tolerance values in Section 5.2 and Section 5.3. The simulation results are provided in Figure 2(a) and Figure 4, validating the robustness of Alt-Diff to the stopping criterion.

---

### Decision · Program_Chairs · 2023-01-20

**Decision:**

Accept: poster

**Justification For Why Not Higher Score:**

 The paper presents a highly methodologically interesting approach, but ultimately the empirical contributions here are a bit minor, and perhaps a bit more narrow in scope that what would be needed for an oral or spotlight.

**Justification For Why Not Lower Score:**

 All reviewers were in agreement that the paper presented a methodologically important and valuable contribution, worthy of acceptance.

**Metareview: Summary, Strengths And Weaknesses:**

Thank you for your submission to ICLR.  The reviewers were in full agreement that this paper proposes an interesting contribution to the field, showing that optimization layers can be solved efficiently using alternating minimization approaches.  This is both an algorithmically and methodologically valuable contribution to the field, which could make such layers more widely used in practice.  Reviewers felt that all questions were addressed in responses, and I'm happy to recommend acceptance with only the recommendation to include these additional results in the final camera-ready version.

**Note From Pc:**

if the above contains the word "oral" or "spotlight" please see: "oral" presentation means -> notable-top-5% and "spotlight" means -> notable-top-25%. As stated in our emails, we are disassociating presentation type from AC recommendations

**Summary Of Ac-Reviewer Meeting:**

N/A